# Rab2A-mediated Golgi-lipid droplet interactions support very-low-density lipoprotein secretion in hepatocytes

Min Xu[1,9], Zi-yue Chen[2,9], Yang Li[3,4,9], Yue Li[1], Ge Guo[1], Rong-zheng Dai[1], Na Ni[1], Jing Tao[3,4], Hong-yu Wang[2], Qiao-li Chen [ID][2], Hua Wang [ID][5], Hong Zhou [ID][1 ✉], Yi-ning Yang [ID][3,4,6,7 ✉], Shuai Chen [ID][2 ✉] & Liang Chen [ID][1,8 ✉]

## Abstract

Lipid droplets (LDs) serve as crucial hubs for lipid trafficking and metabolic regulation through their numerous interactions with various organelles. While the interplay between LDs and the Golgi apparatus has been recognized, their roles and underlying mechanisms remain poorly understood. Here, we reveal the role of Ras-related protein Rab-2A (Rab2A) in mediating LD-Golgi interactions, thereby contributing to very-low-density lipoprotein (VLDL) lipidation and secretion in hepatocytes. Mechanistically, our findings identify a selective interaction between Golgi-localized Rab2A and 17-beta-hydroxysteroid dehydrogenase 13 (HSD17B13) protein residing on LDs. This complex facilitates dynamic organelle communication between the Golgi apparatus and LDs, thus contributing to lipid transfer from LDs to the Golgi apparatus for VLDL2 lipidation and secretion. Attenuation of Rab2A activity via AMP-activated protein kinase (AMPK) suppresses the Rab2A-HSD17B13 complex formation, impairing LD-Golgi interactions and subsequent VLDL secretion. Furthermore, genetic inhibition of Rab2A and HSD17B13 in the liver reduces the serum triglyceride and cholesterol levels. Collectively, this study provides a new perspective on the interactions between the Golgi apparatus and LDs.

**Keywords** Organelle Interactions; Ras-related Protein Rab-2A; 17-Beta-hydroxysteroid Dehydrogenase 13; AMP-activated Protein Kinase; Very-low-density Lipoprotein
**Subject Categories** Metabolism; Organelles

## Introduction

Lipid droplets (LDs), ubiquitous within cells and mainly composed of triglycerides (TG) and cholesterol esters, are enveloped by a phospholipid monolayer, serving diverse functional roles from energy reservoirs to signaling mediators. Increasing evidence has highlighted the critical roles of LD-organelle interactions in lipid trafficking and metabolic regulation (Barbosa et al, 2015; Mathiowetz and Olzmann, 2024; Olzmann and Carvalho, 2019). Notably, systems-level spectral imaging in COS-7 cells has illuminated that a dominant 85% of LDs form interactions with endoplasmic reticulum (ER), establishing it as the primary interaction hub. Subsequently, mitochondria (21%) and Golgi (15%) are identified as significant interaction partners for LDs, with lysosomes or peroxisomes engaging with a smaller fraction (10%), emphasizing a complex organelle network essential for lipid management and metabolic homeostasis (Valm et al, 2017). The Golgi-LD nexus is of particular interest, with studies in murine livers employing Perilipin-2 as a LD marker and GM130 for the Golgi apparatus, revealing extensive interactions, remarkably amplified in hepatocytes of mice on a high-fat diet (Krahmer et al, 2018). Moreover, recent studies have highlighted the potential role of VPS13B in the formation of Golgi-LD communications (Du et al, 2023). PNPLA3, known for its isoleucine-to-methionine substitution at position 148 and its involvement in the progression of fatty liver disease in humans, has now been shown to play important roles in the Golgi-LD dynamics (Sherman et al, 2024). Therefore, it is possible that the communications between Golgi apparatus and LDs play important roles in hepatocytes, but still needs more evidence.

Very-low-density lipoprotein (VLDL), assembled and secreted by the liver, plays critical functions in maintaining lipid homeostasis of serum and liver. VLDLs are highly heterogeneous in size and composition, and typically categorized into two major subfractions (VLDL$_1$ and VLDL$_2$) based on the structural and metabolic characterization. The formation of

[1]College of Life Sciences, Anhui Medical University, 230032 Hefei, China. [2]State Key Laboratory of Pharmaceutical Biotechnology and MOE Key Laboratory of Model Animal for Disease Study, Model Animal Research Center, School of Medicine, Nanjing University, 210061 Nanjing, China. [3]Department of Cardiology, People's Hospital of Xinjiang Uyghur Autonomous Region, 830000 Urumqi, China. [4]Xinjiang Key Laboratory of Cardiovascular Homeostasis and Regeneration Research, 830000 Urumqi, China. [5]Department of Oncology, The First Affiliated Hospital of Anhui Medical University, 230022 Hefei, China. [6]State Key Laboratory of Pathogenesis, Prevention and Treatment of High Incidence Diseases in Central Asia, Xinjiang Medical University, 830000 Urumqi, China. [7]Key Laboratory of Cardiovascular Disease Research, First Affiliated Hospital of Xinjiang Medical University, 830000 Urumqi, China. [8]Department of Critical Care Medicine, The First Affiliated Hospital of Anhui Medical University, 230001 Hefei, China. [9]These authors contributed equally: Min Xu, Zi-yue Chen, Yang Li. ✉E-mail: hzhou@ahmu.edu.cn; yangyn5126@xjrmyy.com; chenshuai@nju.edu.cn; liang-chen@ahmu.edu.cn

mature VLDL$_1$ mainly involves two sequential steps: initially, the Apo B protein undergoes lipidation, resulting in its conversion into smaller and triglyceride-poor VLDL$_2$ within the ER lumen. Subsequently, upon its transport from the ER to the Golgi apparatus, VLDL$_2$ can either be directly secreted or undergo further lipidation to form larger, triglyceride-rich VLDL$_1$ (Boren et al, 2020; Boren et al, 2022; Stillemark-Billton et al, 2005).

Typically, VLDL formation is chiefly regulated by microsomal triglyceride transfer protein (MTP), an ER-resident protein crucial for the initial step of Apo B lipidation (Zhang et al, 2024). Recently, a study revealed that tissue plasminogen activator (tPA) interacts with Apo B, impeding the MTP-Apo B interaction. Conversely, plasminogen activator inhibitor 1(PAI-1) sequesters tPA from Apo B by forming a complex with tPA, thereby facilitating Apo B lipidation and VLDL assembly (Dai et al, 2023). Additionally, other Apo B-interacting proteins such as Transmembrane 6 superfamily member 2 and Lipid transferase CIDEB regulate Apo B lipidation and VLDL formation, thereby modulating VLDL secretion (Li et al, 2020; Smagris et al, 2016; Ye et al, 2009). Subsequently, COP-II-coated vesicles orchestrate VLDL$_2$ transport from the ER to the Golgi apparatus. This process is finely regulated by the secretion-associated Small COPII coat GTPase SAR1B, Surfeit locus protein 4, Transmembrane protein 41B, Small leucine-rich protein 1 (SMLR1), and other associated proteins (van Zwol et al, 2024). Disruption of this finely-tuned mechanism predominantly leads to lipid accumulation within the ER, accompanied by a significant reduction in lipid contents within the Golgi apparatus, thereby impeding subsequent VLDL secretion (Huang et al, 2021; van Zwol et al, 2022; Wang et al, 2021). However, compared with the studies upon Apo B lipidation and its subsequent delivery to the Golgi apparatus, the underlying mechanisms that regulate lipid incorporation into Golgi apparatus and then VLDL$_2$ lipidation remain largely unexplored, while it is widely accepted that cytosolic TG within LDs are the major source for Apo B lipidation (Francone et al, 1989).

Rab GTPase, a family of approximately 60 members in mammalian cells, plays important roles in cellular vesicle trafficking and metabolic homeostasis (Gilleron et al, 2019; Stenmark, 2009). Specifically, Ras-related protein Rab-5 (Zeigerer et al, 2015), Rab-24 (Seitz et al, 2019), Rab-18 (Pulido et al, 2011; Xu et al, 2018), Rab-30 (Smith et al, 2024), and Rab-8A (Chen et al, 2017b; Ouyang et al, 2023) have been demonstrated to have pivotal functions in lipid and glucose metabolism. Recently, our study indicates that hepatocyte Ras-related protein Rab-2A (Rab2A) regulates the serum TG and cholesterol levels with unknown mechanism (Chen et al, 2022). Here, our research contributes to this domain by demonstrating that Rab2A, functioning as a key small G protein positioned on Golgi apparatus, facilitates Golgi-LD interactions in hepatocytes via binding with 17-beta-hydroxysteroid dehydrogenase 13 (HSD17B13), a resident protein on LDs. Rab2A deficiency inhibits these interactions, resulting in decreased TG and cholesterol levels in the Golgi apparatus, which are transported from LDs, and subsequently affecting VLDL$_2$ lipidation and then VLDL secretion in hepatocytes.

# Results

## Deletion of hepatic Ras-related protein Rab-2A (Rab2A) diminishes serum triglyceride and cholesterol levels

To delineate the complicated relationship between hepatic Rab2A and serum lipid levels, we generated a mouse model with hepatocytes-specific Rab2A deletion, termed LCK, with their littermates served as controls and designated as Flox (Appendix Fig. S1). Our findings discovered that hepatic Rab2A deletion markedly reduced serum triglyceride (TG) levels (by ~28.9% under "Random feed" condition and 37.9% under "Fasted" condition) (Fig. 1A) and total cholesterol (TC) levels (by ~41.7% under "Random feed" condition and 48.5% under "Fasted" condition) (Fig. 1B), particularly TG within the very low-density lipoprotein (VLDL) particles (Fig. 1C) and TC within the high-density lipoprotein (HDL) particles (Fig. 1D). Conversely, TG and TC levels in liver (Appendix Fig. S2A,B) exhibited mild corresponding accumulation in the LCK mice, especially under the "Random feed" condition.

Lipoproteins primarily encompass TG, TC, and apolipoproteins, including Apo B-100, Apo B-48, Apo-E, Apo-AI, and Apo-CIII. We evaluated their secretion levels in serum and found them comparable in both genotypes (Fig. 1E,F). Subsequently, we examined the expression levels of apolipoproteins and several lipid transporters in the liver. Apo B levels, comprising Apo B-100 and Apo B-48, were notably increased in the LCK mice (Appendix Fig. S2C,D). Apart from Apo B, protein levels including Apo-E, Apo-AI, Microsomal triglyceride transfer protein (MTP), Low-density lipoprotein receptor (LDLR), and CD36 were all similar in the livers of Flox and LCK mice (Appendix Fig. S2C,D).

Building upon the observed relationship between Rab2A and serum lipid under both "Random feed" and "Fasted" conditions, we further investigated whether Rab2A deficiency in hepatocytes could also confer resistance against hyperlipidemia induced by high-fat-high-cholesterol diet (HFHCD). Remarkably, LCK mice exhibited pronounced rescue in serum TG levels (by ~29.8%) (Fig. 1G) and TC levels (by ~53.4%) (Fig. 1H), particularly evident in TG within the VLDL particles (Fig. 1I) and TC within the HDL particles (Fig. 1J), compared to the similar patterns of hepatic TG and TC (Appendix Fig. S2E,F). In contrast, subsequent detection of apolipoproteins uncovered marked accumulation of Apo B-100 and Apo B-48 proteins in the serum (Fig. 1K,L) and liver (Appendix Fig. S2G,H) of LCK mice.

In summary, our comprehensive analysis indicates that hepatic Rab2A deletion significantly diminishes serum TG and TC levels, but not apolipoproteins levels, suggesting that Rab2A deficiency may attenuate the degree of lipoproteins lipidation.

## Absence of Rab2A hinders the lipidation of VLDL$_2$

In light of the profound impacts of hepatic Rab2A on serum lipid homeostasis, we delved into the mechanistic underpinnings. Our primary objective was to discern whether Rab2A deficiency primarily influenced lipid uptake or secretion. Initially, we evaluated fatty acid absorption rates in mice and primary hepatocytes, observing comparable TG absorption rates between two genotypes (Appendix Fig. S3A–C). We further analyzed the lipid-lowering effects of Rab2A deficiency in LDLR knockout mice, a model characterized by hyperlipidemia due to impaired hepatic lipid uptake (Appendix Fig. S3D). The results demonstrated that Rab2A inhibition still dramatically reduced serum TG levels (by ~26.7%) and TC levels (by ~32.5%) (Appendix Fig. S3E,F), with a notable reduction in the VLDL particles contents (Appendix Fig. S3G,H). These observations suggest that lipid uptake in the liver does not primarily account for the lipid-lowering effects of Rab2A inhibition.

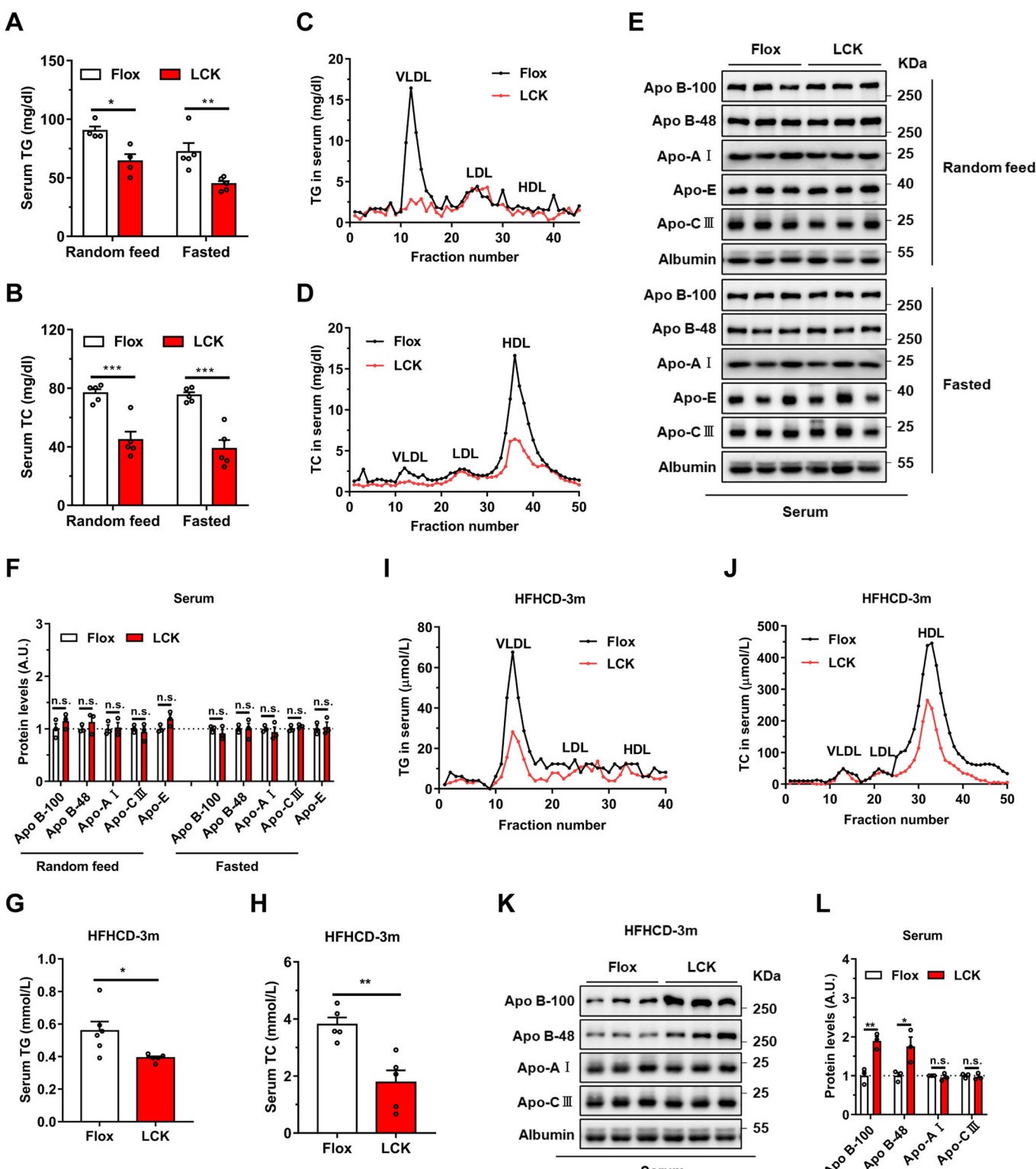

To ascertain the impact of Rab2A on lipid secretion dynamics, we initially assessed lipid secretion rates in Flox and LCK mice using tyloxapol, a lipoprotein lipase inhibitor. Our findings revealed that Rab2A deficiency potentially reduced VLDL lipidation rates in both male and female mice, as evidenced by decreased TG level (Figs. 2A and EV1A,B), with no significant changes in the apolipoproteins (Fig. 2B,C). This suppression of VLDL lipidation level was also validated in the primary hepatocytes (Fig. EV1C,D). To advance our understanding of Rab2A's integral role in VLDL secretion, we employed adeno-associated virus-mediated *Apob* knockdown

Figure 1. Deletion of Ras-related protein Rab-2A (Rab2A) decreases serum triglyceride and cholesterol levels.

(A–F) Serum samples were obtained from Flox and LCK mice under "Random feed" and "Fasted" conditions, followed by subsequent experiments (Male, $n = 5$ mice per group). Serum triglyceride (TG) levels (A) (Random feed, $P = 0.0111$; Fasted, $P = 0.0089$) and their distribution in lipoproteins (C), serum total cholesterol (TC) levels (B) (Random feed, $P = 0.0009$; Fasted, $P = 0.0003$) and their distribution in lipoproteins (D) were analyzed. Then, apolipoprotein secretion levels in serum, such as Apo B-100, Apo B-48, Apo-E, Apo-AI, and Apo-CIII were evaluated (E), with grayscale quantification of the respective proteins, where the value in Flox samples was normalized to 1 (Male, $n = 3$ mice per group) (F). (G–L) Flox and LCK mice were fed with a high-fat-high-cholesterol diet (HFHCD) for three months, followed by the collection of serum samples for experiments (Male, $n = 6$ vs. 5 mice). This encompassed TG levels (G) ($P = 0.0302$) and detailed distributions in lipoproteins (I), TC levels (H) ($P = 0.0033$) and detailed distributions in lipoproteins (J), as well as apolipoproteins in serum samples (K). The expression levels of apolipoproteins were quantified, normalizing Flox samples to 1 (Male, $n = 3$ mice per group) (L) (Apo B-100, $p = 0.0053$; Apo B-48, $P = 0.0407$). Data information: Data in (A, B, F, G, H, L) are presented as mean ± SEM. Circles in (A, B, F, G, H, L) correspond to individual mice. $P$ values in (A, B, F, G, H, L) were determined using unpaired two-tailed Student's $t$-test. n.s. indicates no significant difference ($P > 0.05$); *$P < 0.05$; **$P < 0.01$. VLDL very low-density lipoprotein, LDL low-density lipoprotein, HDL high-density lipoprotein. Source data are available online for this figure.

experiments in both genotypes (Fig. EV1E,F). *Apob* knockdown in the Flox mice led to a substantial decrease in TG (by 59.6%) and TC levels (by 78.4%), whereas in the LCK mice, the reductions were more moderate (only 26.7% for TG and 50.0% for TC) (Fig. EV1G,H). Notably, Apo B suppression aligned the serum lipid profiles of both genotypes, irrespective of Rab2A expression (Fig. EV1G,H). These compelling findings suggest that Rab2A functions downstream of Apo B, playing a crucial role in the secretion of VLDL.

Purification and fractionation of endoplasmic reticulum (ER) and Golgi compartments enabled the assessment of VLDL lipidation levels through Apo B-48 immunoblotting (Li et al, 2012). Subsequently, Golgi apparatus and ER fractions from liver tissues were isolated and purified through sucrose gradient density centrifugation (Appendix Fig. S4). Our investigations revealed that while Apo B lipidation remained consistent in the ER, Rab2A deficiency notably impaired VLDL$_2$ lipidation and increased its density within the Golgi apparatus (Fig. 2D–F). Subsequent analyses of serum lipoproteins' density disclosed a shift towards higher-density lipoproteins in the LCK mice (Fig. 2G,H). Moreover, evaluation of VLDL size demonstrated that Rab2A deletion predisposes LCK mice to secrete smaller-diameter VLDL compared to their wild-type counterparts (Fig. 2I,J).

Collectively, these findings elucidate that Rab2A deficiency in hepatocytes selectively disrupts VLDL$_2$ lipidation within the Golgi apparatus, culminating in the secretion of VLDL particles with diminished lipid contents.

## Rab2A orchestrates Golgi-lipid droplet (LD) interactions

Elucidating the precise subcellular positioning of Rab2A within hepatocytes is pivotal for unraveling its role in VLDL$_2$ lipidation. Previous studies have demonstrated that Rab2A is primarily situated at Golgi apparatus, as evidenced by staining with endogenously genomic-labeled Rab2A (Götz et al, 2021; Lund et al, 2018) and exogenously overexpressed Rab2A (Ding et al, 2019; Gillingham et al, 2014). Our findings corroborated these observations, showing that GTP-bound Rab2A predominantly localized to the Golgi apparatus rather than the ER or ER-Golgi intermediate compartment (ERGIC) compartment, as confirmed by co-localization studies of exogenously expressed Rab2A in Huh7 cells (Appendix Fig. S5). Further analysis with endogenous Rab2A also confirmed its predominant localization to the Golgi apparatus in Huh7 cells, potentially suggesting that Rab2A mainly existed in the GTP-bound form (Fig. 3A). In contrast to the concentrated Golgi morphology observed in Huh7 cells, primary hepatocytes exhibit a more dispersed Golgi distribution (Sherman et al, 2024) (Figs. 3B and EV2B; Appendix Fig. S6A). Moreover, the localization of Rab2A in primary hepatocytes appears more complex. While Rab2A still remained primarily in the Golgi apparatus, signals outside the Golgi apparatus were also observed, which may reflect the GDP-bound form of Rab2A at lysosomes (Ding et al, 2019; Yin et al, 2017) (Fig. 3B and EV5E). Additionally, studies in fly larval ventral nerve cords (Götz et al, 2021) and Hela-S3 cells (Aizawa and Fukuda, 2015) have highlighted Rab2A's role in maintaining Golgi apparatus integrity. Our investigations extended these findings to primary hepatocytes, indicating that Rab2A deficiency had a negligible impact on the structural integrity of the Golgi apparatus (Appendix Fig. S6), suggesting varied roles of Rab2A in different cell types.

To delineate the mechanistic contribution of Golgi-anchored Rab2A to VLDL$_2$ lipidation, we isolated and purified Golgi apparatus and ER compartments from the livers of Flox and LCK mice. Subsequent quantitative evaluations were conducted to measure TG and TC levels, as well as the abundance of Apo B protein within these compartments. Our studies revealed that Rab2A deficiency led to a significant reduction in TG and TC levels in the Golgi compartment but not within the ER fractions (Fig. 3C–F). Conversely, the Apo B protein exhibited an obvious enrichment in the Golgi compartment, without notable alterations in the ER (Fig. 3G). These findings led us to hypothesize that Rab2A may act as an important molecule, facilitating lipid transfer from LDs to the Golgi apparatus by mediating their interactions.

Initial observations in Huh7 cells and primary hepatocytes confirmed a pronounced interplay between Golgi membranes-resident Rab2A and LDs, indicating frequent communications (Fig. EV2). Subsequently, we analyzed the potential contact points between the Golgi apparatus and LDs, and the underlying role of Rab2A. Under oleic acid (OA) incubation, an average of approximately 22 points between the Golgi apparatus and LDs was observed per primary hepatocyte (Fig. 3H,I), a frequency higher than that reported in a previous study within the Hep3B cell line (~5 points) (Sherman et al, 2024). Rab2A deficiency significantly reduced these potential contact points to about 17 (Fig. 3H,I). The finding was further corroborated by a LDs pull-down assay (Appendix Fig. S7), which elucidated that Rab2A suppression significantly decreased the interface between LDs and Golgi, as evidenced by the reduced levels of Golgi markers in Rab2A-depleted samples (Fig. 3J). This effect was specific to the Golgi-LD interface, with Rab2A inhibition showing a comparable impact on the interactions between ER and LDs (Fig. 3J).

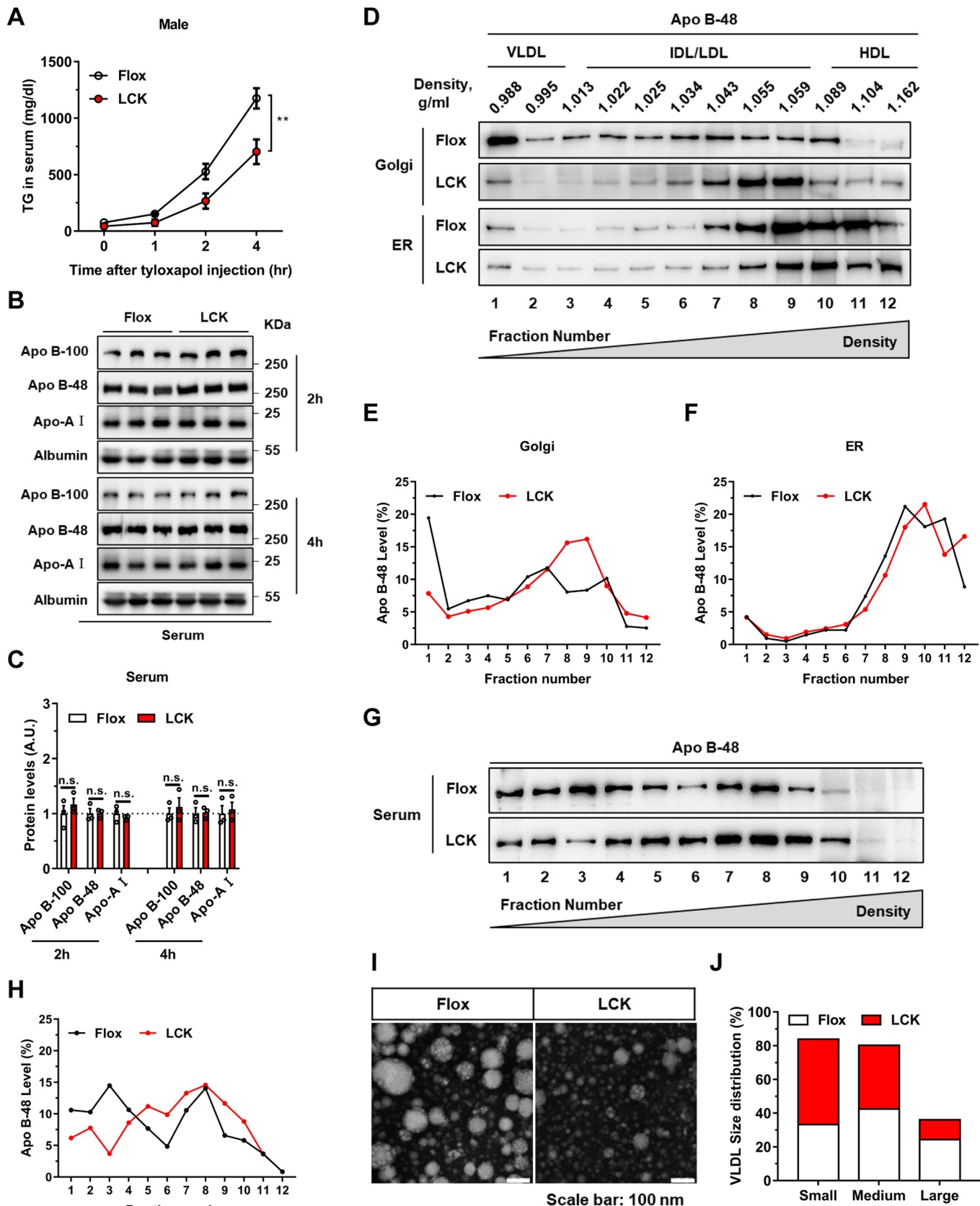

**Figure 2.   Absence of Rab2A hinders the lipidation process of VLDL₂.**

(A–C) VLDL-TG secretion in male mice were assessed through a tyloxapol injection assay (A) ($n = 6$ vs. 5 mice) ($P = 0.0058$). Subsequent analysis and quantification of apolipoprotein secretion levels in serum collected at two hours and four hours following tyloxapol injection (Male, $n = 3$ mice per group) (B, C). (D–H) Quantification of the degree of lipoproteins lipidation within the endoplasmic reticulum (ER), Golgi apparatus (Golgi), and serum of Flox and LCK mice using the sucrose density gradient centrifugation method. Protein imprinting of Apo B-48 (D, G) and subsequent statistical results (E, F, H) elucidated the distribution and proportion of lipoproteins, with fraction 1 representing the top layer of sucrose density and fraction 12 corresponding to the bottom layer. The experiments were replicated twice with similar pattern (Male, $n = 1$ mouse per group). (I, J) The size of VLDL secreted into the serum of Flox and LCK mice (Male, $n = 3$ mice per group) was analyzed by negative staining and transmission electron microscope (TEM) imaging. Representative images (I) and statistical data (J) are presented, the number of particles is more than 200. Data information: Data in (A, C) are presented as mean ± SEM. Circles in (C) correspond to individual mice. $P$ value in (A) was determined using two-way ANOVA. $P$ values in (C) were determined using unpaired two-tailed Student's $t$ test. n.s. indicates no significant difference ($P > 0.05$); **$P < 0.01$. Source data are available online for this figure.

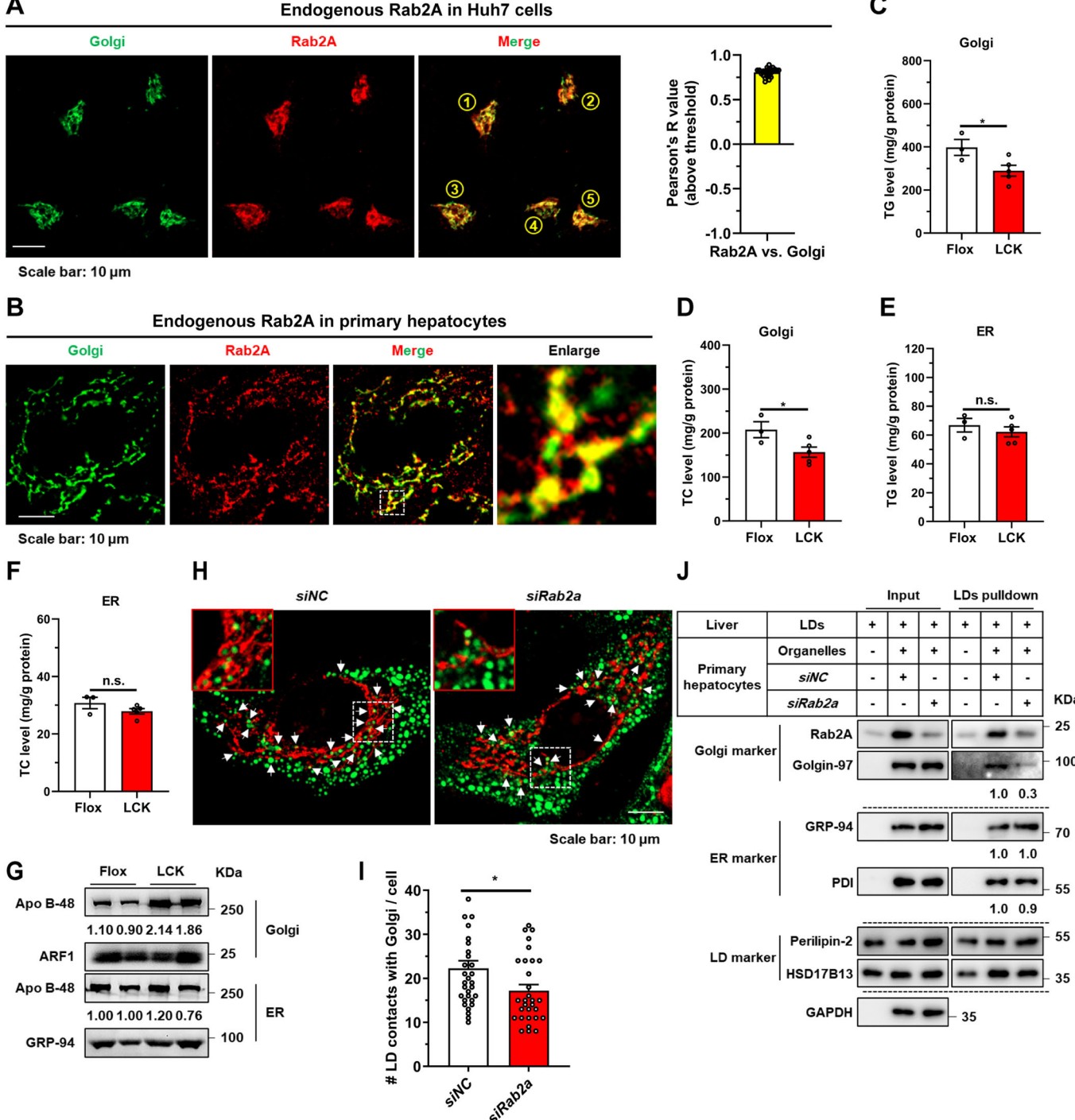

**Figure 3. Rab2A orchestrates Golgi-Lipid droplet (LD) interactions.**

(A) The subcellular localization of Rab2A was examined in Huh7 cells. Endogenous Rab2A was stained with a primary antibody, and Golgi apparatus was labeled using a primary antibody targeting GM130. Representative images from 5 cells are shown (left), and the percentage of colocalization was quantified with Pearson's R value ($n = 24$ cells) (right). (B) The subcellular localization of Rab2A was also confirmed in mouse primary hepatocytes. Endogenous Rab2A was stained with a primary antibody, and Golgi apparatus was labeled with primary antibodies against GM130. Representative images are shown. (C-G) Sucrose density gradient centrifugation facilitated the isolation and purification of Golgi and ER compartments from livers of Flox and LCK mice (Male, $n = 3$ vs. 5 mice). TG and TC levels were precisely assessed in Golgi fractions (C, D) (TG, $P = 0.0445$; TC, $P = 0.0457$) and ER fractions (E, F). Apo B-48 protein levels were also analyzed in Golgi and ER fractions with ADP-ribosylation factor 1 (ARF1) or GRP-94 as the internal control (Male, $n = 2$ mice per group) (G). (H, I) Mouse primary hepatocytes were isolated, transfected with corresponding Rab2A-siRNA, incubated with 100 μM oleic acid (OA) for 16 h, and then stained with a primary antibody against GM130 (Red). LDs were labeled with Bodipy (Green). Representative images are shown and white arrows indicate potential contacts (H). Statistical data are presented (I) ($n = 31$ vs. 30 cells) ($P = 0.0274$). (J) Inhibition of Rab2A attenuated Golgi-LD interfacing, as evidenced by LDs pulldown assay. Purified LDs, isolated from the livers of wild-type mice, were incubated with organelle clusters from primary hepatocytes with or without Rab2A deficiency. The assay was performed twice with similar conclusion and representative results are shown. Data information: Data in (A, C–F, I) are presented as mean ± SEM. Circles in (A, I) correspond to individual cell. Circles in (C–F) correspond to individual mice. $P$ values in (C–F, I) were determined using unpaired two-tailed Student's $t$ test. n.s. indicates no significant difference ($P > 0.05$); *$P < 0.05$. Source data are available online for this figure.

Taken together, our findings suggest that Rab2A orchestrates the interaction between the Golgi apparatus and LDs, critically modulating the lipid transfer from LDs to the Golgi apparatus, thus pivotal in the lipidation of VLDL$_2$.

## 17-beta-hydroxysteroid dehydrogenase 13 (HSD17B13), an LD-localized protein, binds with Rab2A to mediate Golgi-LD interactions and VLDL secretion

In pursuit of unraveling the underlying molecular mechanisms by which Rab2A influences Golgi-LD interactions, our initial approach entailed co-immunoprecipitation assays aimed at identifying potential Rab2A binding partners. HSD17B13, a protein localized to LDs (Fig. 4A), emerged as a key candidate (Abul-Husn et al, 2018; Ma et al, 2019; Ma et al, 2020; Su et al, 2014; Wang et al, 2022). The binding between Rab2A and HSD17B13 was robustly confirmed across liver tissue samples, cell lines, and through in vitro assays (Figs. 4B and EV3A–C). Moreover, this binding was dependent on the Rab2A's activity and Golgi localization (Fig. EV3F; Appendix Figs. S8A–C and S9), as well as the LD localization of HSD17B13 (Appendix Fig. S8D,E). Subsequent analysis utilizing immunofluorescence further revealed partial colocalization between endogenously expressed Rab2A and HSD17B13 in primary hepatocytes (Fig. 4C), with ~16.2% of the HSD17B13 signal contacting with Rab2A (Fig. EV3D,E).

Exploring the functional implications of HSD17B13 in Golgi-LD communications and VLDL secretion, we observed that inhibition of HSD17B13 in primary hepatocytes resulted in a decreased number of Golgi-LD interactions, subsequently decreasing TG secretion (Fig. 4D–G), a finding further validated by the liver-specific knockdown of HSD17B13 using adeno-associated viruses (Appendix Fig. S10A; Fig. 4H). Remarkably, HSD17B13 deficiency significantly reduced Golgi-LD interfacing, while leaving ER-LD interactions partially increased, as demonstrated by the LD-pulldown assay (Fig. 4I). Following HSD17B13 inhibition, serum TG and TC levels were attenuated (Fig. 4J,K), with a paradoxical increase in liver TG levels, likely due to compensatory mechanisms (Appendix Fig. S10B,C). Additionally, analysis of serum apolipoproteins revealed that HSD17B13 knockdown diminished the secretion of Apo B-48, in conjunction with disrupted lipid secretion (Appendix Fig. S10D,E), suggesting alternative mechanisms through which HSD17B13 influenced VLDL secretion beyond its binding with Rab2A.

Further exploration through molecular mapping pinpointed critical amino acids within Rab2A, specifically residues 32–42, as crucial for its binding with HSD17B13 (Appendix Fig. S11). Mutation of these residues abrogated the complex formation (Fig. EV4A) and significantly impaired TG secretion (Fig. EV4D,E), without altering Rab2A localization and activity (Fig. EV4B,C). These results underscore the indispensable role of Rab2A-HSD17B13 interaction in modulating TG secretion.

Collectively, our comprehensive dataset elucidates a novel regulatory axis wherein the proteins binding between Golgi-localized Rab2A and LD-associated HSD17B13 mediates Golgi-LD interactions, thereby facilitating VLDL secretion.

## AMP-activated protein kinase (AMPK) signaling attenuates Rab2A activity and its role in Golgi-LD interactions and VLDL secretion

To deepen our understanding of Rab2A's role in the Golgi-LD interactions and subsequent VLDL secretion, we focused on the regulatory effects of AMPK on Rab2A activity (Chen et al, 2022). Fasting, a physiological condition known to markedly increase AMPK activity (Zong et al, 2019), has been extensively studied for its inhibitory effects on VLDL secretion (Cheng et al, 2016; Rai et al, 2017). Our data confirmed these findings, demonstrating that fasting significantly inhibited VLDL secretion (Appendix Fig. S12). More importantly, fasting also reduced Rab2A activity and its Golgi localization upon AMPK activation (Fig. 5A,B; Appendix Fig. S13), suggesting Rab2A's involvement in VLDL secretion under normal feeding conditions.

To further investigate Rab2A activity modulation, we employed A769662, a well-known agonist of AMPK, which effectively reduced Rab2A activity (Figs. 5C and EV5A,D,F), leading to a notable relocation from the Golgi apparatus to the cytosol, potentially to the lysosomes (Fig. EV5B,C,E). Subsequently, A769662 stimulation significantly disrupted Rab2A-HSD17B13 complex (Fig. EV5G), and then attenuating Golgi-LD interactions (Fig. 5D–F). Additionally, we validated that AMPK activation via A769662 dramatically reduced TG secretion by about 40% in wild-type, but not in Rab2A-deficient, primary hepatocytes, demonstrating that AMPK suppresses TG secretion primarily through Rab2A (Fig. 5G).

In summary, our findings further highlight the crucial role of Rab2A, likely modulated by AMPK signaling, in binding with HSD17B13 to facilitate dynamic Golgi-LD interactions and promote VLDL secretion.

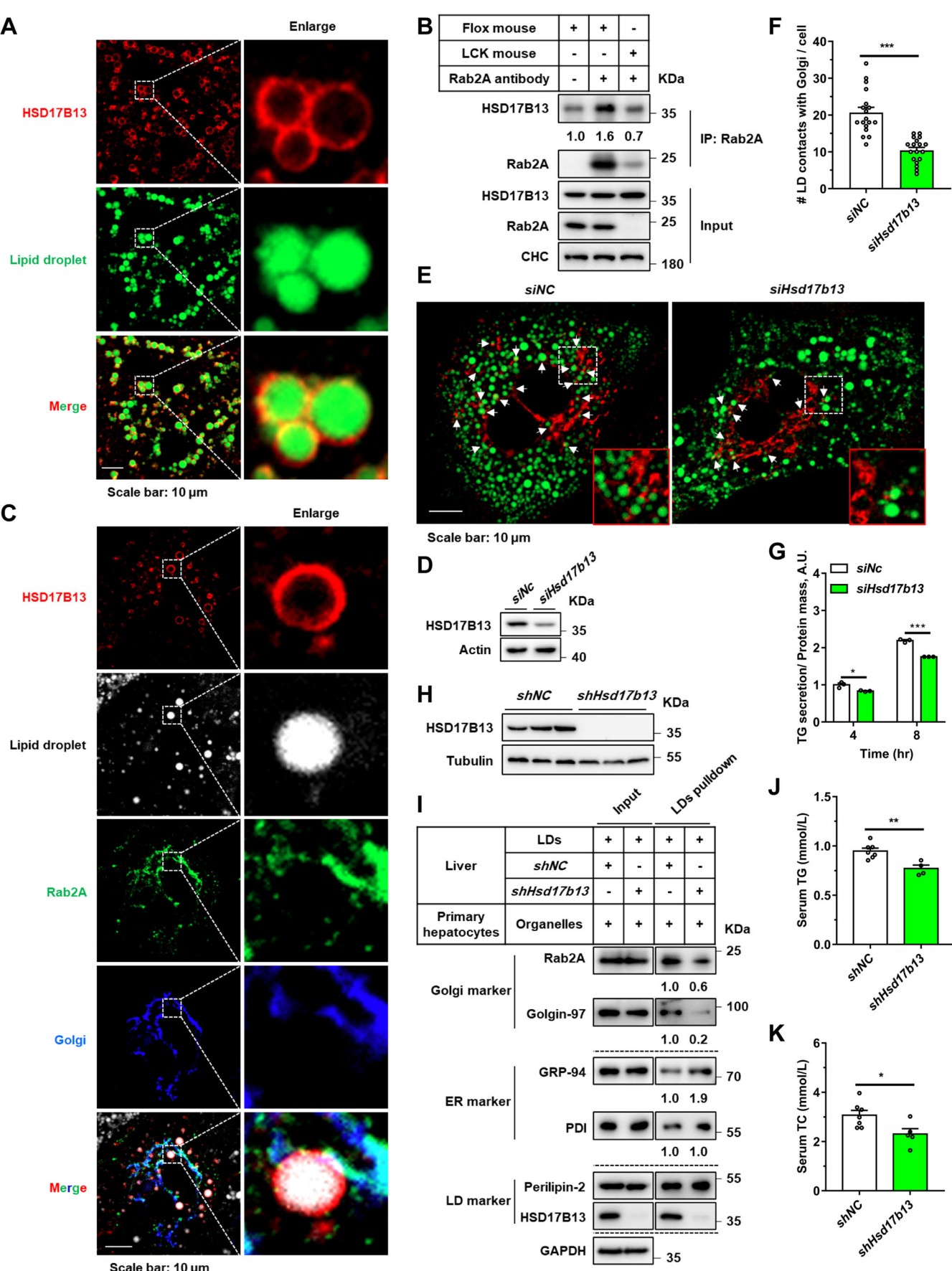

**Figure 4. 17-beta-hydroxysteroid dehydrogenase 13 (HSD17B13), an LD-localized protein, binds with Rab2A to mediate Golgi-LD interactions and VLDL secretion.**

(A) Mouse primary hepatocytes were isolated, cultured and then stained with a primary antibody against endogenous HSD17B13, with LDs labeled with Bodipy. Representative images are shown. (B) The liver samples from Flox and LCK mice were processed and incubated with Rab2A against primary antibody, facilitating the enrichment of the interaction between Rab2A and HSD17B13 using protein A/G affinity beads, and subsequently visualized via Western blotting. The assay was performed twice with similar results. (C) Mouse primary hepatocytes were isolated, cultured and then stained with primary antibodies against endogenous Rab2A and HSD17B13. The Golgi apparatus was labeled by overexpressing BFP-RCAS1, while LDs were visualized using HCS LipidTOX. Representative images are presented. (D–G) Mouse primary hepatocytes were isolated, transfected with corresponding siRNA targeting endogenous HSD17B13. Knockdown efficiency was validated by Western blotting (D). Hepatocytes were incubated with 100 μM oleic acid (OA) for 16 h then fixed and stained for the Golgi apparatus using a primary antibody against GM130 (Red) and LDs using Bodipy (Green). Representative images are shown, with white arrows indicating potential contacts (E). Statistical data are presented (F) ($n = 18$ cells per group) ($P < 0.0001$). TG secretion levels were quantified and normalized with cellular total protein content (G) (4 h, $P = 0.0209$; 8 h, $P < 0.0001$). (H–K) In vivo inhibition of HSD17B13 was achieved in wild-type mouse livers using AAV2/8-shRNA virus (Male, $n = 7$ vs. 5 mice). Several experiments were conducted to explore the impact of HSD17B13 on organelle communications and VLDL secretion. The effectiveness of knockdown (H), Golgi-LD interfacing (I), TG and TC levels in serum (J, K) (TG, $P = 0.0041$; TC, $P = 0.0267$) were systematically assessed. The assay in "I" was performed twice with similar conclusion and representative results are shown. Data information: Data in (F, G, J, K) are presented as mean ± SEM. Circles in (F) correspond to individual cell. Circles in (G) correspond to individual test. Circles in (J, K) correspond to individual mice. $P$ values in (F, G, J, K) were determined using unpaired two-tailed Student's $t$ test. *$P < 0.05$; **$P < 0.01$; ***$P < 0.001$. Source data are available online for this figure.

## Discussion

In this study, our findings demonstrate that Ras-related protein Rab-2A (Rab2A) acts as a pivotal regulator at the Golgi apparatus, facilitating Golgi-Lipid droplets (LDs) interactions and thereby contributing to very-low-density-lipoprotein (VLDL) secretion in hepatocytes. Specifically, we demonstrate that Golgi-localized Rab2A binds with 17-beta-hydroxysteroid dehydrogenase 13 (HSD17B13), a LD-resident protein. This interaction, potentially regulated by AMP-activated protein kinase (AMPK) signaling, orchestrates organelle communications between the Golgi apparatus and LDs, and then enhancing lipid transport from LDs into the Golgi apparatus. Finally, the lipids within the Golgi apparatus promote the further lipidation of VLDL$_2$, leading to the formation of mature VLDL$_1$ (Fig. 6).

Here, our investigation into Rab2A-HSD17B13 complex presents a novel mechanism for exploring proteins involved in the Golgi-LD interactions. Given Rab2A's broad expression across various cell types, its role in Golgi-LD interactions in other tissues warrants further investigation. Moreover, our findings suggest that other Golgi-localized small GTPases, such as Ras-related protein Rab-1, Rab2B and Rab-6, could be promising candidates for further research into inter-organelle communications. Additionally, AMPK, a key energy sensor, regulates cellular catabolism via substrates phosphorylation (Hardie et al, 2012; Herzig and Shaw, 2018). Our findings indicate that Rab2A activity is suppressed by AMPK, though the upstream GTPase-activating proteins (GAPs) and guanine exchange factors (GEFs) for Rab2A remain unidentified, necessitating further investigation into this regulatory axis.

Lipid trafficking between LDs and other organelles is generally mediated by two principal mechanisms: membrane fusion facilitated by Soluble N-ethylmaleimide-sensitive factor attachment protein receptors (SNAREs) (Jahn and Scheller, 2006), and channel transport driven by lipid transport proteins (LTPs) (Reinisch and Prinz, 2021). For instance, lipid transport between the ER and LDs is partially characterized, with SNARE proteins (Syntaxin18, Use1, BNIP1) playing pivotal roles in lipid flux from the ER to LDs (Xu et al, 2018). In parallel, VPS13, a member of the LTP family, has been identified as a regulator of lipid trafficking across the ER, mitochondria, and LDs nexus (Gao and Yang, 2018; Kumar et al, 2018; Leonzino et al, 2021; Wang et al, 2021). Although our findings indicate lipid transport between the Golgi apparatus and

LDs, further studies are needed to fully elucidate the mechanisms underlying this process.

The assembly, transport, and lipidation of VLDL have been extensively studied, yet the specific mechanisms governing VLDL$_2$ lipidation remain less well understood. Previous investigations have shed light on the Golgi-based lipidation of VLDL$_2$, focusing on the roles of Lipid transferase CIDEB (CIDEB) and Perilipin-2 (Li et al, 2012; Ye et al, 2009). Hepatic CIDEB deficiency dramatically diminishes triglyceride (TG) levels and VLDL$_2$ lipidation within the Golgi apparatus, a modulation speculated to be influenced by increased Perilipin-2 expression (Li et al, 2012). However, the precise mechanisms by which the ubiquitously expressed CIDEB and the LD-specific Perilipin-2 regulate VLDL$_2$ lipidation remain elusive. Our current findings reveal that the hepatic deficiency of Golgi-localized Rab2A markedly reduces TG and total cholesterol (TC) levels only in the Golgi apparatus (Fig. 3C,D), specifically impairing Golgi-centric VLDL$_2$ lipidation (Fig. 2D–F) through disrupted Golgi-LD interactions (Fig. 3H–J). Our study offers new insights into the regulation of VLDL$_2$ lipidation based on Golgi-LD interactions.

HSD17B13, predominantly expressed in hepatocytes, occupies a critical niche in liver disease etiology (Lindén and Romeo, 2023). Investigations in human cohorts have demonstrated that loss-of-function mutations in *HSD17B13* gene confer protection against liver inflammation and fibrosis, paradoxically correlating with enhanced lipid deposition (Abul-Husn et al, 2018; Luukkonen et al, 2020; Ma et al, 2019). Subsequent findings indicate that the suppression of pyrimidine catabolism in mice lacking HSD17B13 may underpin the impact of HSD17B13 on the progression of liver fibrosis (Luukkonen et al, 2023). However, murine models have yielded contradictory insights into HSD17B13's role in the hepatic lipid metabolism. While adenovirus-mediated overexpression of human HSD17B13 in mice has been linked to elevated hepatic lipid levels (Su et al, 2014), its genetic deletion has also been associated with exacerbated liver steatosis or macro-vesicular steatosis (Adam et al, 2018; Ma et al, 2021). Our findings contribute to this field by demonstrating that hepatic HSD17B13 deficiency leads to significant TG accumulation, largely due to the impaired Golgi-LD interactions and subsequent disruptions in VLDL secretion (Fig. 4D–K), which may partially explain the phenotypes observed in the human studies.

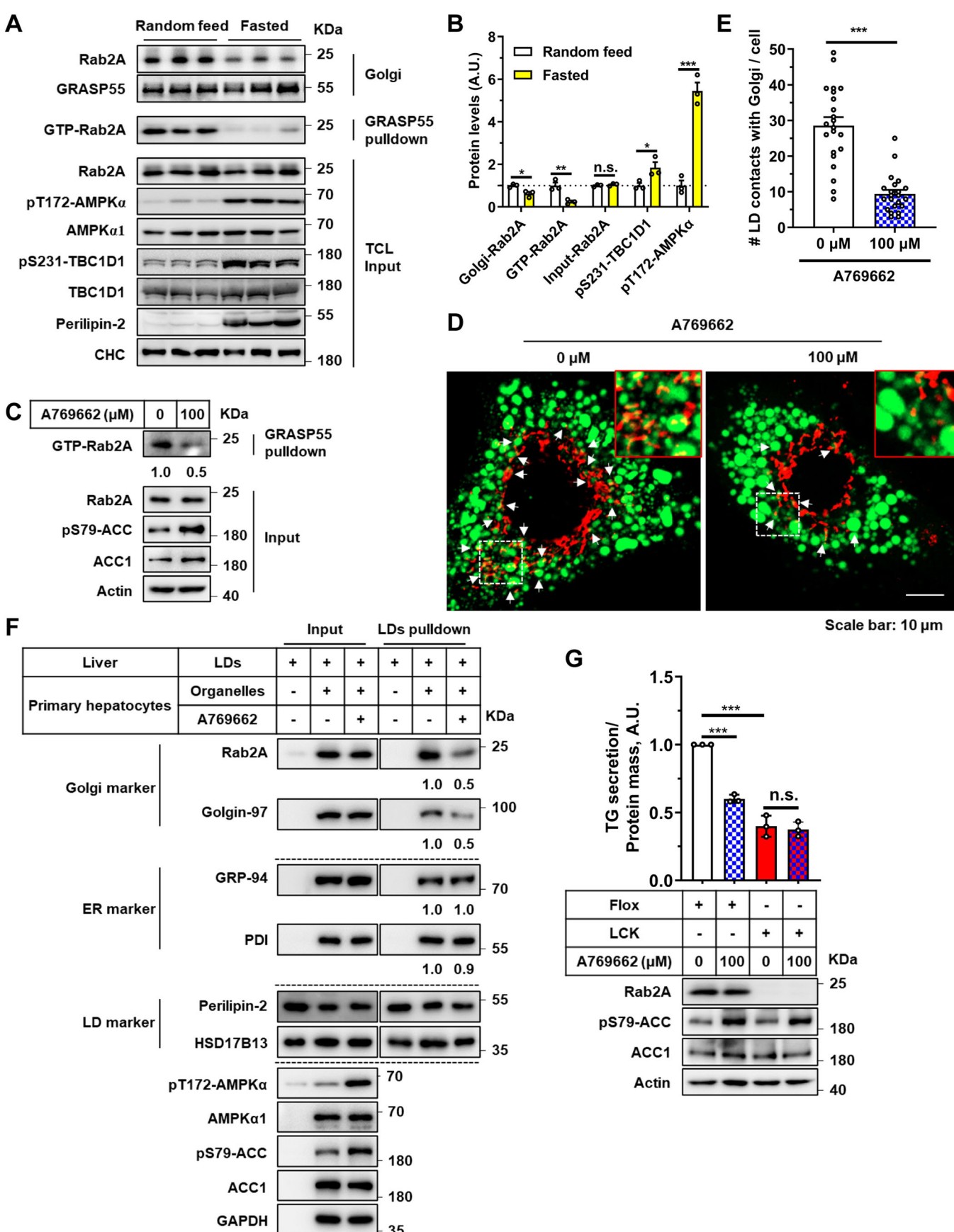

Scale bar: 10 µm

◀ **Figure 5.  AMPK signaling attenuates Rab2A activity and its role in Golgi-LD interactions and VLDL secretion.**

(**A, B**) Rab2A activity and subcellular distribution in liver samples after overnight fasting were analyzed using the GRASP55 pulldown assay and sucrose density gradient centrifugation assay (Male, $n = 3$ mice per group) (**A**), with corresponding statistical analyses presented (**B**) (Golgi-Rab2A, $P = 0.0194$; GTP-Rab2A, $P = 0.0089$; pS231-TBC1D1, $P = 0.0454$; pT172-AMPKα, $P = 0.0006$), where levels in "Random feed" state samples were normalized to 1. The assay was performed twice with similar results. (**C–F**) Primary hepatocytes were stimulated with A769662 (100 μM), an AMPK agonist, for 4 h in DMEM medium without fetal bovine serum (FBS), followed by a series of assays. Rab2A activity was primarily evaluated using the GRASP55 pulldown assay (**C**). Potential Golgi-LDs contact points were then quantified by staining Golgi apparatus with a primary antibody against GM130 (Red) and LDs with Bodipy (Green). Representative images are shown, with white arrows indicating potential contacts (**D**). Statistical data are presented (**E**) ($n = 22$ cells per group) ($P < 0.0001$). Subsequently, the LDs pulldown assay was performed by incubating purified LDs from the livers of wild-type mice with organelle clusters from primary hepatocytes, with or without A769662 stimulation (**F**). The assay in "F" was performed twice, yielding similar results, with representative data shown. (**G**) Primary hepatocytes isolated from Flox and LCK mice were cultured and treated with A769662 (100 μM) for 6 h in DMEM medium without FBS. TG secretion levels were measured and quantified (A769662 stimulation in Flox hepatocytes (0 μM vs. 100 μM), $P < 0.0001$; Flox vs. LCK in hepatocytes without A769662 stimulation (0 μM), $P = 0.0002$), and cell lysates were prepared for Western blot analysis ($n = 3$ per group). Data information: Data in (**B, E, G**) are presented as mean ± SEM. Circles in (**B**) correspond to individual mice. Circles in (**E**) correspond to individual cell. Circles in (**G**) correspond to individual assay. $P$ values in (**B, E, G**) were determined using unpaired two-tailed Student's $t$ test. n.s. indicates no significant difference ($P > 0.05$); *$P < 0.05$; **$P < 0.01$; ***$P < 0.001$. Source data are available online for this figure.

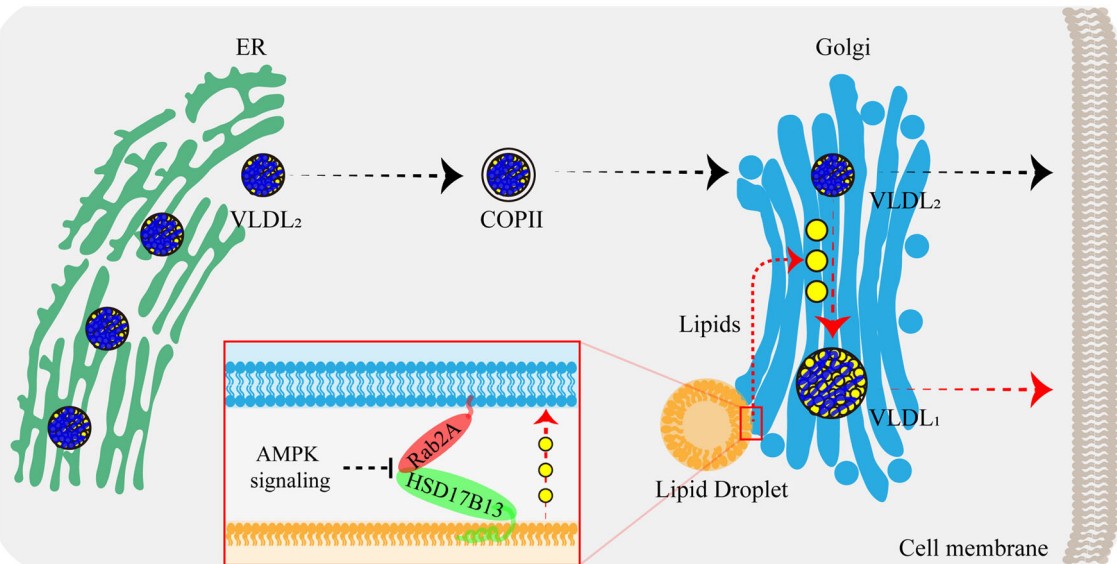

**Figure 6.  The graphical abstract describes the pivotal role of Rab2A in mediating Golgi-LD interactions and subsequent VLDL secretion in hepatocytes.**

VLDL$_2$, assembled in the ER, is transported to the Golgi via COP-II vesicles for further processing. Our studies reveal that VLDL$_2$ lipidation primarily occurs within the Golgi apparatus, where lipids are transferred from LDs, leading to the formation of mature VLDL$_1$. Both VLDL$_1$ and VLDL$_2$ are secreted from the Golgi apparatus into the serum via vesicular systems. Mechanistically, Golgi-localized Rab2A binds with HSD17B13, an LD-resident protein, orchestrating organelle interactions between the Golgi apparatus and LDs, thereby enhancing lipid transport and VLDL secretion. Additionally, AMPK signaling can potentially attenuate these processes by inhibiting Rab2A activity.

# Methods

### Reagents and tools table

| Reagent/resource | Reference or source | Identifier or catalog number |
|---|---|---|
| **Experimental models: cell lines** | | |
| Human: HEK293T | Cell Resource Center, Chinese Academy of Medical Sciences and Peking Union Medical College | 1101HUM-PUMC000091 |
| Human: Huh7 | Cell Resource Center, Chinese Academy of Medical Sciences and Peking Union Medical College | 1101HUM-PUMC000679 |

| Reagent/resource | Reference or source | Identifier or catalog number |
|---|---|---|
| Mouse: WT(C57BL/6) primary hepatocytes | This paper | N/A |
| Mouse: Rab2A-Flox primary hepatocytes | This paper | N/A |
| Mouse: Rab2A-KO primary hepatocytes | This paper | N/A |
| **Experimental models: organisms/strains** | | |
| Mouse: C57BL/6 J: WT | Gempharmatech | N000013 |
| Mouse: *Rab2a*-flox/flox | Gempharmatech | T018874 |
| Mouse: *Ldlr*-KO | Gempharmatech | T001464 |
| Mouse: *Alb*-iCre | Gempharmatech | T003814 |
| **Recombinant DNA** | | |

| Reagent/resource | Reference or source | Identifier or catalog number |
|---|---|---|
| pHAGE-3*Flag-Rab2A (Mouse) | This paper | N/A |
| pCDNA3-3*Flag-Rab2A (Mouse) | This paper | N/A |
| pCDNA5-MYC-BirA*-Rab2A (Mouse) | This paper | N/A |
| pCDNA5-RCAS1-BFP (Mouse) | This paper | N/A |
| pCDNA3-EGFP-Rab2A (Mouse) | This paper | N/A |
| pGEX-6p-1-GST-GRASP55 (Mouse) | This paper | N/A |
| pGEX-6p-1-GST-HSD17B13 (Mouse) | This paper | N/A |
| pCDNA5-HSD17B13-YFP (Mouse) | This paper | N/A |
| pCDNA3-5*MYC-HSD17B13 (Mouse) | This paper | N/A |
| pCDNA3-HSD17B13-mCherry (Mouse) | This paper | N/A |
| pCDNA3-ERGIC-53-mCherry (Mouse) | This paper | N/A |
| **Antibodies** | | |
| Rabbit anti-Apo B | Proteintech | 20578-1-AP |
| Mouse anti-Apo-E | Proteintech | 66830-1-Ig |
| Mouse anti-Apo-AI | Proteintech | 66206-1-Ig |
| Mouse anti-Apo-CIII | Santa Cruz | sc-293227 |
| Mouse anti-Microsomal triglyceride transfer protein large subunit (MTP) | Santa Cruz | sc-515742 |
| Rabbit anti CD36 | Cell Signaling | 14347 |
| Mouse anti-Clathrin heavy chain (CHC) | Santa Cruz | sc-12734 |
| Mouse anti Albumin | Proteintech | 66051-1-Ig |
| Mouse anti-Ras-related protein Rab-2A (Rab2A) | Proteintech | 67501-1-Ig |
| Mouse anti-Tubulin | DUONENG-BIO | AB0178801 |
| Rabbit anti-pT172-AMPK subunit alpha (AMPKα) | Cell Signaling | 2535 |
| Rabbit anti-pS79-ACC | Cell Signaling | 3661 |
| Mouse anti-AMPK subunit alpha-1 (AMPKα1) | Proteintech | 66536-1-Ig |
| Mouse anti-ACC1 | Proteintech | 67373-1-Ig |
| Rabbit anti-pS231-TBC1 domain family member 1 (TBC1D1) | Millipore | 07-2268 |
| Rabbit anti-TBC1 domain family member 1 (TBC1D1) | Cell Signaling | 4629 |
| Rabbit anti-Perilipin-2 | Proteintech | 15294-1-AP |

| Reagent/resource | Reference or source | Identifier or catalog number |
|---|---|---|
| Mouse anti-ADP-ribosylation factor 1 (ARF1) | Santa Cruz | sc-53168 |
| Rabbit anti-GRP-94 | Proteintech | 14700-1-AP |
| Rabbit anti-GM130 | Proteintech | 11308-1-AP |
| Mouse anti-Golgin-97 | Cell Signaling | 97537 |
| Rabbit anti-PDI | Cell Signaling | 3501 |
| Rabbit anti-Perilipin-3 | Proteintech | 10694-1-AP |
| Rabbit anti-17-beta-hydroxysteroid dehydrogenase 13 (HSD17B13) | Abcam | ab122036 |
| Mouse anti-GRASP55 | Santa Cruz | sc-271840 |
| Mouse anti-Flag | Sigma | F1804 |
| Rabbit anti-Flag | Proteintech | 20543-1-AP |
| Mouse anti-MYC | ATCC | CRL-1729 |
| Rabbit anti-MYC | Proteintech | 16286-1-AP |
| Mouse anti-GST | Santa Cruz | sc-53909 |
| Mouse anti-Actin | Zen-bioscience | 200068-8F10 |
| Rabbit anti-GAPDH | Proteintech | 10494-1-AP |
| Mouse anti-Lamin-B1 | Proteintech | 66095-1-Ig |
| Rabbit anti-EGFP | Bao-liang Song, Wuhan university | N/A |
| Rabbit anti-LDLR | Bao-liang Song, Wuhan university | N/A |
| Peroxidase-AffiniPure Goat Anti-Mouse IgG (H + L) | Jackson ImmunoResearch Laboratories | 115-035-003 |
| Peroxidase-AffiniPure Goat Anti-Rabbit IgG (H + L) | Jackson ImmunoResearch Laboratories | 111-035-003 |
| **Oligonucleotides and other sequence-based reagents** | | |
| Primers for Genotyping, see below | This paper | N/A |
| shRNA targeting sequence: Mouse APOB | This paper | N/A |
| shRNA targeting sequence: Mouse HSD17B13 | This paper | N/A |
| siRNA targeting sequence: Mouse Rab2A | This paper | N/A |
| siRNA targeting sequence: Mouse HSD17B13 | This paper | N/A |
| **Chemicals, enzymes and other reagents** | | |
| Anti-Flag affinity beads | Smart-Lifesciences (China) | SA042005 |
| anti-MYC affinity beads | Smart-Lifesciences (China) | SA065005 |
| anti-EGFP affinity beads | Smart-Lifesciences (China) | SA070005 |
| Streptavidin beads 6FF | Smart-Lifesciences (China) | SA021001 |
| Glutathione-Sepharose 4B beads | GE Healthcare | 17-0756-01 |

| Reagent/resource | Reference or source | Identifier or catalog number |
|---|---|---|
| Linear polyethylenimine (LPEI) | PolyScience | 24314-2 |
| Tyloxapol | Sigma | T0307 |
| Iodixanol | Merck | D1556 |
| Lipofectamine-3000 | Thermo Fisher | 100022052 |
| A769662 | MCE | HY-50662 |
| Atglistain | MCE | HY-15859 |
| Digitonin | MCE | HY-N4000 |
| HSL-IN-3 | MCE | HY-15859 |
| LipidTOX | Thermo Fisher | H34477 |
| LysoTracker | Invitrogen | L7528 |
| BODIPY | Thermo Fisher | D3922 |
| DAPI | Biosharp | BL105A |
| BODIPY™ FL C12 | Thermo Fisher | D3822 |
| Free glycerol reagent | Sigma | F6428 |
| Glycerol | Sigma | G7793 |
| TG kit | Nanjing Jiancheng Bioengineering Institute, China | A110-1-1 |
| TG kit | Wako | 290-63701 |
| TC kit | Nanjing Jiancheng Bioengineering Institute, China | A111-2-1 |
| TC kit | Wako | 294-65801 |
| **Software** | | |
| GraphPad Prism V5 | GraphPad | https://www.graphpad.com/features |
| ImageJ software | NIH | https://imagej.nih.gov/ij/ |
| ZEN | ZEISS | https://www.zeiss.com/microscopy/zh/home.html |
| Photoshop | Adobe | https://www.adobe.com/cn/ |
| Tanon-5200 | Tanon | http://en.biotanon.com/ |
| **Other** | | |
| Rab2A- bound proteins detected by mass spectrometry | This paper | N/A |

## Animals

Male C57BL/6JGpt mice (Strain No. N000013), *Rab2a*-flox/flox mice (Strain No. T018874), *Ldlr*-KO mice (Strain No. T001464), and *Alb*-iCre mice (Strain No. T003814) were generated and procured from Gempharmatech (Nanjing, China). Rab2a hepatocytes-specific knockout mice were generated by crossing *Rab2a*-flox/flox mice with *Alb*-iCre mice.

while *Ldlr*-KO/ *Rab2a*-flox/flox mice were produced by crossing *Ldlr*-KO mice with *Rab2a*-flox/flox mice. Genotyping primers were listed: *Rab2a*-flox/flox mice: 5′-CACTCACAGACACATTCCCACACA-3′ and 5′-AGCAAGCCTTGGTCTTTCCAAC-3′; *Alb*-iCre mice: 5′-TGGATGCCACCTCTGATGAAGTC-3′ and 5′-TCCTGGCATCTGTCAGAGTTCTCC-3′; *Ldlr*-KO mice: 5′-CTCCCAGGATGACTTCCGAT-3′ and 5′-CGCAGTGCTCCTCATCTGAC -3′. Mice with diet-induced hyperlipidemia were fed with either Western diet (WD) (No. D12079B, Research Diets) or high-fat-high-cholesterol diet (HFHCD) (No. D09100310, Research Diets) starting from 8 weeks.

Adeno-associated virus serotype 2/8 (AAV2/8) -mediated gene knockdown in the mouse liver was conducted as previously described (Chen et al, 2022). The AAV2/8 viruses pAAV-U6-shRNA (*NC*)-Cbh-EGFP-WPRE, pAAV-U6-shRNA (*Apob*) -Cbh-EGFP-WPRE and pAAV-U6-shRNA (*Hsd17b13*) -Cbh-EGFP-WPRE were obtained from Obio Technology (Shanghai, China). The sequences of shRNA-*Apob* and shRNA-*Hsd17b13* were as follows: shRNA-*Apob* (Cheng et al, 2016): (5′-ACCGCAGACAAGCACCTGGAAATTCTCGAGAATTTCCAGGTGCTTGTCTGCTTTTTTG-3′ and 5′-CTAGCAAAA AAGCAGACAAGCACCTGGAAATTCTCGAGAATTTCCAGGTGCTTGTCTGC-3′); shRNA-*Hsd17b13* (Wang et al, 2022): (5′-ACCGGCGTCATCATCTACTCCTACCCTCGAGGGTAGGAGTAGATGATGACGCTTTTTTG-3′ and 5′-CTAGCAAAAAAGCGTCATCATCTACTCCTACCCTCGAG GGTAGGAGTAGATGATGACGC-3′).

Male mice were randomized into groups for each experiment. Eight-week-old wild-type C57BL/6 J mice received intravenous injections of $5 \times 10^{11}$ vg AAV2/8 virus via the tail using a 29-gauge insulin syringe (BD). All assays were conducted two weeks post-AAV2/8 virus injection. Mice were housed in a pathogen-free environment with a 12-hour light/12-hour dark cycle and had ad libitum access to water and food. All animal breeding, husbandry, care, and use procedures adhered to the guidelines outlined by the Ethics Committees of Anhui Medical University (Approval number LLSC20200327 and LLSC20241078).

## Cell culture, transfection, knockdown and plasmids

Human embryonic kidney HEK293T cells, human liver carcinoma Huh7 cells were sourced from the Cell Resource Center, Chinese Academy of Medical Sciences and Peking Union Medical College (China). Generally, transient transfection followed established protocols (Chen et al, 2022), where cell seeding occurred on day 0, plasmid transfection with linear polyethylenimine (LPEI) was performed on day 1, and detailed experiments were conducted on day 3.

The plasmids in this study were constructed using standard molecular cloning techniques, incorporating site mutations and sequence truncations via quick-change mutagenesis. Key plasmids are listed in "Reagents and Tools" table.

Isolation of mouse primary hepatocytes was conducted as previously described (Chen et al, 2017a; Chen et al, 2016). Plasmids transfection and siRNA knockdown (*siHsd17b13*: GCGTCAT-CATCTACTCCTACC; *siRab2a*: GCCTATCTCTTCAAGTA-CATC) in primary hepatocytes utilized Lipo3000 reagents or viruses (Lenti-virus or AAV2/8-virus). Lentiviral transfection was constructed in our laboratory. The AAV2/8 viruses pAAV-TBG-Rab2A-Flag-P2A-GFP and pAAV-TBG-Rab2A (Δ032-042)-Flag-P2A-GFP were provided by the ChuangRui Bio (Lian Yungang, China). Cells were cultured at 37 °C with 5% CO2 in DMEM

(Biological Industries) supplemented with 100 units/ml penicillin, 100 µg/ml streptomycin sulphate (Thermo Fisher), and 10% fetal bovine serum (FBS, Biological Industries).

## Immunoprecipitation, immunoblotting and antibodies

Proteins immunoprecipitation (IP) and immunoblotting were conducted in accordance with established protocols (Chen et al, 2022). Briefly, tissue or cell samples were promptly collected and homogenized in the respective RIPA buffer or IP buffer containing proteinase inhibitors. For immunoprecipitation assay, quantified samples were primarily incubated with immune-beads for several hours, followed by removal of non-specific binding proteins through washing. Subsequently, the aliquots were quantified, subjected to SDS-PAGE, transferred to PVDF, and incubated with relevant antibodies. Western blotting signals were then captured using an autoradiography machine (Tanon-5200). Quantification of protein levels was performed using Image J (National Institutes of Health, https://imagej.nih.gov/ij/) and normalized to the internal reference. The primary antibodies for LDLR and EGFP were generously gifted by Professor Song (Zhou et al, 2023). Detailed information regarding primary antibodies and secondary antibodies are listed in "Reagents and Tools" table.

## Rab2A binding protein immunoprecipitation and LC-MS/MS

Liver protein samples (5 mg) were extracted from wild-type mice using IP buffer containing proteinase inhibitors. The primary antibody against Rab2A (5 µl) was incubated with the liver samples overnight, and binding proteins were collected with protein A/G affinity beads (No. L-1004, Biolinkedin, China). Subsequently, the beads were prepared for mass spectrometry at Bioprofile Company (Shanghai, China).

In Brief, the bound proteins were extracted from IP beads using lysis buffer (4% SDS, 100 mM DTT, 100 mM Tris-HCl pH 8.0). The IP beads samples were boiled for 3 min and further ultrasonicated. Undissolved beads were removed by centrifugation at $16{,}000 \times g$ for 15 min, and the supernatant containing proteins were collected. Finally, the protein suspension was digested with 2 µg trypsin (Promega) overnight at 37 °C. The peptides were collected by centrifugation at $16{,}000 \times g$ for 15 min and desalted with C18 StageTip for further LC-MS analysis.

LC-MS/MS experiments were performed on a Q Exactive Plus mass spectrometer coupled to Easy nLC1200 (Thermo Scientific). Peptides were initially loaded onto a trap column in buffer A (0.1% Formic acid in water). Reverse-phase high-performance liquid chromatography (RP-HPLC) separation was carried out using a self-packed column at a flow rate of 300 nl/min. The RP-HPLC mobile phase A was 0.1% formic acid in water, and B was 0.1% formic acid in 95% acetonitrile. The gradient was set as following: 2–4% buffer B from 0 min to 2 min, 4% to 30% buffer B from 2 min to 47 min, 30% to 45% buffer B from 47 min to 52 min, 45% to 90% buffer B from 52 min to 54 min, 90% buffer B was maintained until 60 min. MS data was acquired using a data-dependent top20 method dynamically choosing the most abundant precursor ions from the survey scan (350–1800 $m/z$) for HCD fragmentation. The full MS scans were acquired at a resolution of 70,000 at $m/z$ 200, and 15,000 at $m/z$ 200 for MS/MS scan. The maximum injection time was set to for 50 ms for MS and 25 ms for MS/MS. Normalized collision energy was 28 and the isolation window was set to 1.6 $m/z$. Dynamic exclusion duration was 30 s.

The MS data were analyzed using MaxQuant software version 1.6.1.0. MS data were searched against the UniProtKB Mus musculus database. The database search results were filtered and exported with <1% false discovery rate (FDR) at peptide-spectrum-matched level, and protein level, respectively. The summary of Rab2A-specific binding proteins is presented in Appendix Table S1.

## Immunofluorescence staining and imaging

Huh7 cells were transfected with related plasmids expressing Rab2A, HSD17B13, RCAS1, ERGIC-53, each tagged with a fluorescent protein epitope. Endogenous Rab2A was stained with a primary antibody (Proteintech technology, 67501-1-Ig, 1:100). Golgi apparatus labeling was achieved using a GM130 primary antibody (Proteintech technology, 11308-1-AP, 1:200) or BFP-RCAS1. ER-Golgi intermediate compartment (ERGIC) was labeled with mCherry-ERGIC-53 fluorescent protein. Endoplasmic reticulum (ER) was visualized using a KDEL expression plasmid provided by Professor Baoliang Song. Lipid droplets were stained with BODIPY ((4,4-Difluoro-1,3,5,7,8-Pentamethyl-4-Bora-3a,4a-Diaza-s-Indacene), No. D3922, Thermo Fisher) or HCS LipidTOX (No. H34477, Thermo Fisher), and lysosome was labeled with lysoTracker (Invitrogen, LysoTracker™ Red DND-99, L7528).

In mouse primary hepatocytes, the Golgi was labeled with GM130 (Proteintech technology, 11308-1-AP, 1:100) or Golgin-97 (Cell signaling technology, 13192, 1:100) primary antibodies, lipid droplets were visualized using BODIPY or HCS LipidTOX, and cellular nuclei were stained with DAPI (No. BL105A, Biosharp). Endogenous Rab2A localization was evaluated with a mouse anti-Rab2A primary antibody (Proteintech technology, 67501-1-Ig, 1:100), and endogenous HSD17B13 localization was detected using a rabbit anti-HSD17B13 primary antibody (Thermo Fisher, PA5-109834, 1:20).

For fatty acid uptake in mouse primary hepatocytes, the cultured cells were pretreated in serum-free medium for 4 h, followed by treatment with 2.5 µM bovine serum albumin (BSA)-conjugated fatty acid (BODIPY™ FL C12, Thermo Fisher, D3822) for 1 h according to the standard protocol (Hao et al, 2020).

Generally, standard procedures, including washing, fixation (4% paraformaldehyde (PFA) for 30 min at room temperature or ice-cold methanol for 15 min at 4 °C), quenching (50 mM NH₄Cl for 15 min), permeabilization (0.1% Triton X-100 for 10 min or 50 µg/ml digitonin for 5 min), blocking (3% BSA for 30 min), staining primary antibodies, secondary antibodies, and mounting, were performed. Subsequently, images were captured using a Zeiss confocal microscope (LSM800 and LSM980). The representative results are presented and the corresponding quantification was performed using Image J (National Institutes of Health, https://imagej.nih.gov/ij/).

## Measurement of lipids level in liver and serum

Triglyceride (TG) and total cholesterol (TC) quantification were performed on frozen liver and serum samples as previously described (Chen et al, 2022). For liver samples, lipids extraction involved weighing, homogenizing, saponifying, and extracting. TG levels in liver were determined using the free glycerol reagent (No. F6428, Sigma-Aldrich) with glycerol (No. G7793, Sigma-Aldrich)

as the standard. TC levels in the liver was measured using a LabAssay Cholesterol kit (No. 294-65801; Wako Chemicals USA, Inc.) following the standard protocol. The quantification of parameters in serum samples were determined using TG kits (No. A110-1-1, Nanjing Jiancheng Bioengineering Institute, China; No. 290-63701, Wako Chemicals USA, Inc.) and TC kits (No. A111-2-1, Nanjing Jiancheng Bioengineering Institute, China; No. 294-65801, Wako Chemicals USA, Inc.).

## Fast protein liquid chromatography (FPLC)

Lipoprotein profiling analysis was conducted as previously described (Zhang et al, 2019). In brief, 500 μL of pooled serum was loaded onto a Superdex 200 Increase gel filtration column (10/300 GL, GE Healthcare, No. 28-9909-44) using an ÄKTA puro FPLC system, separated at a flow rate of 0.3 ml/min in standard phosphate-buffered saline buffer supplemented with 5 mM EDTA, and 200 μl per fractions were collected for TG and TC detection (Fraction 30-Fraction 95).

## Transmission electron microscope

Primary hepatocytes, as detailed in the preceding methodology, were immersed in 1 ml of 2.5% (v/v) glutaraldehyde for 4–5 h at 4 degrees Celsius. Subsequently, the samples underwent three 15-minute washes with 0.1 M phosphate buffer (PB buffer, pH 7.0). Following this, fixation was performed using a 1% osmium tetroxide solution for 1–2 h. After meticulous removal of excess osmium tetroxide, an additional three 15-minute washes with PB buffer were carried out. Dehydration was systematically performed using a series of ethanol solutions (30%, 50%, 70%, 80%, 90%, and 95%) for 15 min each, followed by two treatments with anhydrous ethanol for 20 min each. Finally, the samples underwent a 20-minute treatment with pure acetone. A one-hour treatment with a mixture of embedding agent and acetone (v/v = 1/1) preceded a three-hour treatment with a mixture of embedding agent and acetone (v/v = 3/1). Subsequently, the samples were treated overnight with pure embedding agent. Following infiltration overnight, the samples were embedded and heated at 70 °C overnight to achieve well-embedded specimens. Using a LEICA EM UC7 ultramicrotome, 70 nm sections were obtained. These sections were stained with a lead citrate solution and a 50% ethanol-saturated solution of uranyl acetate for 5 min each, before being observed under a transmission electron microscope.

## VLDL-TG secretion in mice and primary hepatocytes

For the in-vivo VLDL-TG secretion assay, mice were intraperitoneal injected with tyloxapol (an inhibitor of lipoprotein lipase (LPL)) at a dose of 500 mg/kg following a 16-h fasting. Subsequently, the tail vein serum was collected at 0, 1, 2, and 4 h, individually, and subjected to TG detection. In the ex-vivo VLDL-TG secretion assay with primary hepatocytes, cultured hepatocytes following plasmids or siRNA transfection were initially incubated with 200 μM BSA-conjugated oleate acid containing ATGL and HSL inhibitors for 2 h. The cells were then washed with phosphate-buffered saline (PBS) and then incubated with Opti-MEN adding 1% fatty acid-free BSA. The medium was consecutive collected for TG measurement at indicated hours.

## Fatty acids absorption in mice

Mice were subjected to a 4-h fasting period and subsequently administered olive oil (6 μl/g) via oral gavage, following the established protocol (Chen et al, 2017b). The tail vein serum was then collected at the specified time points for the detection of TG.

## Subcellular fractionation and lipids measurement

Subcellular fractionation was conducted according to established procedures (Li et al, 2012). In brief, Fresh liver samples were promptly harvested and homogenized in ice-cold lysis buffer (10 mM HEPES, pH 7.4; 150 mM sucrose; 0.5 mM DTT; and 1× cocktail inhibitors) using a loosely fitted Dounce homogenizer for ~20 cycles. The homogenates were then centrifuged at 1900 × g for 10 min, and the resulting supernatant underwent ultracentrifugation at 100,000 × g for 90 min using a Beckman SW60Ti rotor. The lipids fraction was floated at the top, and the pellet was collected, resuspended in 800 μl of an 8.58% sucrose solution, and then loaded onto the top of a sucrose density gradient with layers at the following concentrations (from the top to bottom): 20% (160 μl), 30% (480 μl), 35% (800 μl), 40% (800 μl), 45% (480 μl), 50% (320 μl), and 60% (160 μl) sucrose. Following ultracentrifugation at 43,900 rpm for 18 h in a Beckman SW60Ti rotor, 12 fractions (~324 μl per fraction) were collected from top to bottom. The distribution patterns of the subcellular compartment markers were assessed through Western blotting, utilizing GRP94 and PDI as markers for endoplasmic reticulum (ER) and GM130, Golgin-97 as markers for the Golgi. Golgi fractions (Fraction 3–5) and ER fractions (Fraction 8–12) were combined and centrifugated at 100,000 × g for 90 min. The resulting pellets were then resuspended in lysis buffer (20 mM Tris-HCl, 150 mM NaCl, 1 mM EDTA, 1 mM EGTA, 1% Triton-X100, and protease inhibitors, at pH 7.4). TG and TC levels were measured using kits (TG, No. 290-63701; TC, No. 294-65801, Wako Chemicals), with protein mass serving as the internal control.

## Sucrose gradient separation of Apo B-containing lipoproteins in Golgi, ER and serum

The isolation of Apo B-containing lipoproteins from Golgi, endoplasmic reticulum (ER), and serum was performed as previously described (Li et al, 2012). Briefly, Golgi and ER pellets obtained from the preceding centrifugation step were lysis with a 1.6 ml solution buffer containing 0.1 M sodium carbonate (pH 11.0) and deoxycholic acid (0.025%) for 30 min at room temperature. Subsequently, BSA was added to achieve a final concentration of 5 mg/ml, and the sample was centrifugated at 50,000 rpm for 1 h in a Beckman SW60Ti rotor. The resulting supernatant was adjusted to PH 7.4 with addition of 10% acetic acid, brought to a total volume of 1.8 ml with PBS, and adjusted to a sucrose concentration of 12.5% (w/v). The supernatant was layered in a centrifugation system, composed from top to bottom: PBS (982 μl), sample (1.686 ml), 20% sucrose (666 μl), and 49% sucrose (666 μl). All solutions were supplemented with protease inhibitors. Following centrifugation at 33,500 rpm for 43 h in a Beckman SW60Ti rotor, 12 fractions (each 324 μl) with densities ranging from 0.988 g/ml to 1.162 g/ml were collected from the top of the tube. The density of Lipoproteins in each fraction was calculated by comparing with the weight of water (VLDL, 0.96 g/ml–1.006 g/ml; IDL/LDL, 1.006 g/ml–1.063 g/ml; HDL, 1.063 g/ml–1.21 g/ml). The

percentage of lipoproteins was evaluated through immunoblotting using Apo B as the marker protein.

Serum (200 µl) collected after tyloxapol injection for 4 h was diluted into 1.686 ml with PBS, and also adjusted to a sucrose concentration of 12.5% (w/v). Following the same system and method as described above, 12 fractions were separated after centrifugation and Apo B was also selected as the marker for lipoproteins.

## VLDL isolation and transmission electron microscopy analysis

Serum isolation and transmission electron microscopy analysis were conducted as previously detailed (Wang et al, 2020). In summary, 110 µl of serum collected from mice after tyloxapol injection for 4 h was initially diluted 1:30 with 3.19 ml PBS. Subsequently, 1.0 ml iodixanol (60% density) was added to 3.0 ml mixture to achieve a 15% concentration of iodixanol. The final mixture underwent ultracentrifugation with SW60Ti at $350,000 \times g$ for 3 h. The top 20 µl of very-low-density lipoprotein (VLDL) were collected for negative staining using phosphotungstic acid. Images were captured with a JEOL JEM-1400Plus. The diameter of VLDL particles was quantified by Image J software and subclassified into larger VLDL (60–200 nm), medium VLDL (35–60 nm) and small VLDL (27–35 nm), with analysis conducted on more than 200 VLDL particles (Nikolac, 2014; Wojczynski et al, 2011).

## Lipid droplets pulldown assay

LDs in liver tissues (sh*NC* vs. sh*Hsd17b13* or wild-type) were purified according to a well-constructed protocol with some modifications (Ding et al, 2013). In brief, about 500 mg of liver was homogenized on ice ten times with a loose-fitting Dounce in buffer A solution (20 mM tricine, 250 mM sucrose (pH 7.8) and 0.2 mM PMSF) and then centrifuged at $100 \times g$ for 10 min at 4 °C to remove almost unbroken tissues. Subsequently, the suspensions from liver were disrupted with syringe (22 G) and then centrifugated at $3000 \times g$ for 10 min to remove nuclei, cell debris and unbroken cells. The resulting supernatant (1 ml) were transferred to new Eppendorf tubes for further centrifugation ($2000 \times g$ for 30 min at 4 °C) after being loaded with 200 µl of buffer B solution (20 mM HEPES, 100 mM KCl and 2 mM MgCl2 (pH 7.4)) on the top. LDs were carefully collected from the top band of gradient and washed 3 times with buffer B ($20,000 \times g$ for 5 min at 4 °C).

The organelle pulldown assay was also modified from a previous procedure (Krahmer et al, 2018). Briefly, mouse primary hepatocytes, either with wild-type, Rab2A deficiency, or stimulated with A769662 (100 µM, 4 h), were directly collected and homogenized in lysis buffer (200 mM Tris pH 7.4, 0.5 mM EDTA, 5 mM KCl, 3 mM MgCl$_2$, protease inhibitor, phosphatase inhibitor cocktail), and then centrifugated at $1000 \times g$ for 10 min to remove nuclei, cell debris and unbroken cells. LDs in the cell lysate were further removed by centrifugation at $20,000 \times g$ for 15 min. Subsequently, purified LDs from the upper steps (about 400 µg proteins) were isolated and incubated with the respective cellular organelles mix (about 800 µg proteins) at 37 °C for 1.5 h. Following this incubation, LDs and interacting organelles (ER and Golgi) were collected via centrifugation at $20,000 \times g$ for 15 min, and then LDs on the top band of gradient were washed twice with buffer B ($20,000 \times g$ for 5 min at 4 °C). The results were analyzed using western blotting.

## Statistical analysis

All statistical data are expressed as the mean ± standard error of the mean (s.e.m.). Each experiment was independently replicated at least thrice without specific statements, yielding consistent results. Statistical analyses were conducted using GraphPad Prism 7 (GraphPad Software). Unless otherwise indicated, unpaired two-tailed Student's *t* test and two-way ANOVA were employed for data analysis, and significance was determined at $P < 0.05$.

## Data availability

All the plasmids and specific reagents generated in this study are available upon request. The Source Data for main figures have been uploaded. This study includes no data deposited in external repositories.

The source data of this paper are collected in the following database record: biostudies:S-SCDT-10_1038-S44318-024-00288-x.

## Peer review information

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

## Acknowledgements

We extend our gratitude to Professor Baoliang Song for his generous assistance with the numerous plasmids and antibodies. Our sincere appreciation also goes to Professor Xiuyun Wang for providing invaluable insights into sucrose density gradient centrifugation techniques. Funding for this research was provided by various sources, including the National Natural Science Foundation of China (82000549, 82470614 to LC and 32025019 to SC), the Natural Science Foundation of Anhui Province (2008085MC67 to LC), the Doctoral Start-up Foundation of Anhui Medical University (0810013101 to LC), the Outstanding Youth Science Foundation of the Department of Education of Anhui Province (2023AH030058 to LC), the Technology Innovation Leading Talent Project (2022195955 to Y-NY), and the Ministry of Science and Technology of China (2018YFA0801100 and 2021YFF0702100 to SC and H-YW).

## Author contributions

**Min Xu**: Data curation; Formal analysis; Investigation. **Zi-yue Chen**: Data curation; Formal analysis; Investigation. **Yang Li**: Data curation; Formal analysis; Investigation. **Yue Li**: Investigation. **Ge Guo**: Investigation. **Rong-zheng Dai**: Investigation. **Na Ni**: Investigation. **Jing Tao**: Investigation. **Hong-yu Wang**: Investigation. **Qiao-li Chen**: Investigation. **Hua Wang**: Investigation. **Hong Zhou**: Resources; Supervision; Investigation; Project administration; Writing—review and editing. **Yi-ning Yang**: Resources; Supervision; Funding acquisition; Investigation; Project administration. **Shuai Chen**: Resources; Supervision; Funding acquisition; Investigation; Project administration; Writing—review and editing. **Liang Chen**: Conceptualization; Resources; Data curation; Software; Formal analysis; Supervision; Funding acquisition; Validation; Investigation; Visualization; Methodology; Writing—original draft; Project administration; Writing—review and editing.

Source data underlying figure panels in this paper may have individual authorship assigned. Where available, figure panel/source data authorship is listed in the following database record: biostudies:S-SCDT-10_1038-S44318-024-00288-x.

## Disclosure and competing interests statement

The authors declare no competing interests.

# Expanded View Figures

**Figure EV1.   Evaluating the rates of lipid secretion in Flox and LCK mice.** ▶

(**A**) Triglyceride (TG) secretion in female mice were assessed through a tyloxapol injection assay ($n = 7$ vs. 6 mice) ($P = 0.0021$). (**B**) A visual examination of blood transparency in Flox and LCK mice was conducted four hours post-tyloxapol injection. (**C, D**) The evaluation of the TG secretion in primary hepatocytes involved quantifying the secretion level of TG (**C**) ($P < 0.0001$) and Apo B-48 (**D**), with normalization of Flox samples' secretion level at 1-hour to 1. (**E–H**) The schematic representation outlines the timeline for the *Apob* knockdown assay (**E**), followed by the validation of Apo B-48 proteins levels (**F**) and assessment of serum TG (**G**) (Flox (*shNC* vs. *shApob*), $P < 0.0001$; *shNC* (Flox vs. LCK), $P < 0.0001$; LCK (*shNC* vs. *shApob*), $P = 0.0079$) and TC (**H**) (Flox (*shNC* vs. *shApob*), $P < 0.0001$; *shNC* (Flox vs. LCK), $p = 0.0009$; LCK (*shNC* vs. *shApob*), $P = 0.0034$) levels under "Fasted" condition (Male, $n = 10$ vs. 5 vs. 7 vs. 8 mice). Data information: Data in (**A, C, G, H**) are presented as mean ± SEM. Circles in (**G, H**) correspond to individual mice. *P* values in (**A, C**) were determined using two-way ANOVA. *P* values in (**G, H**) were determined using unpaired two-tailed Student's *t*-test. n.s. indicates no significant difference ($P > 0.05$), ** indicates $P < 0.01$; *** indicates $P < 0.001$.

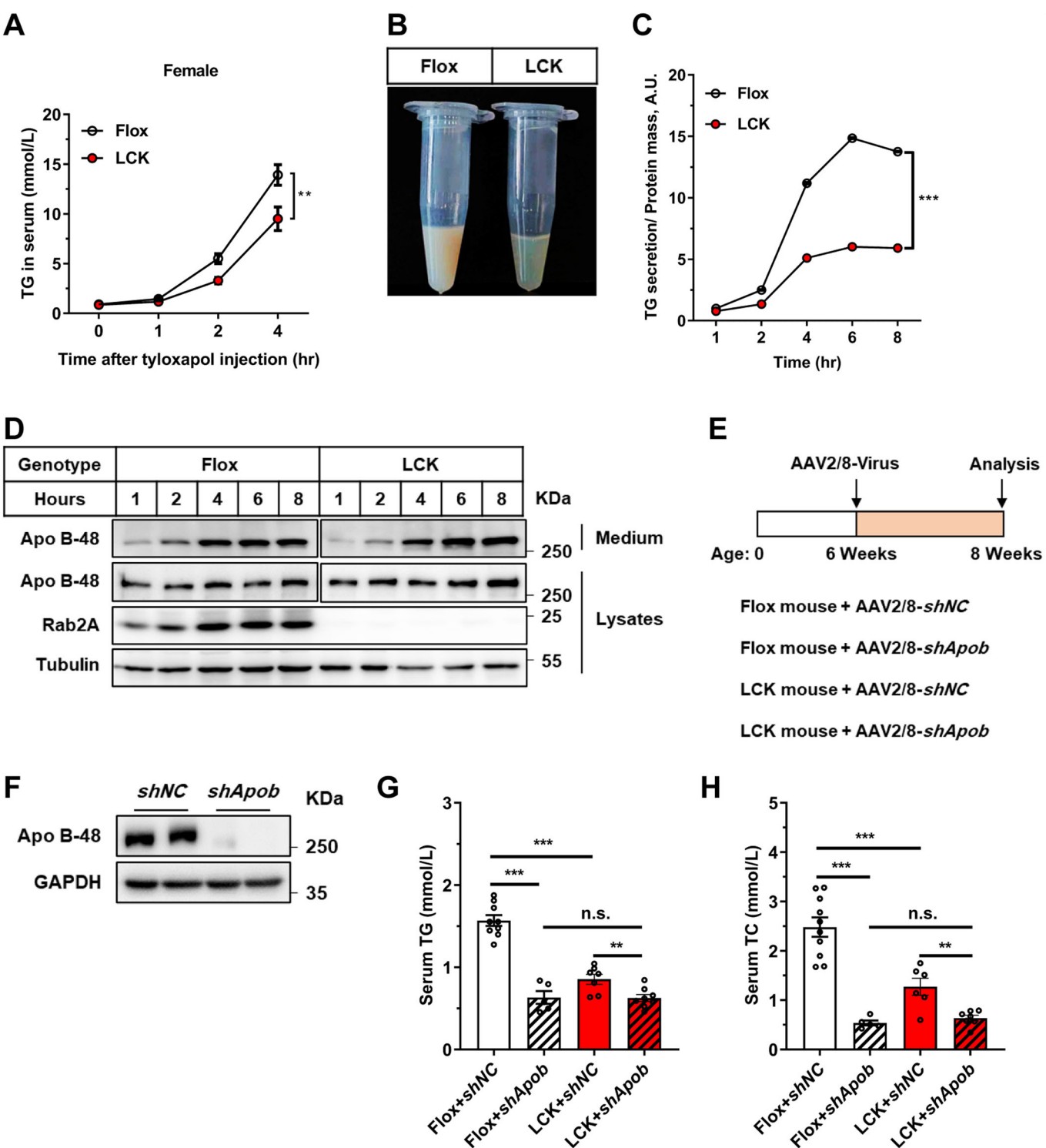

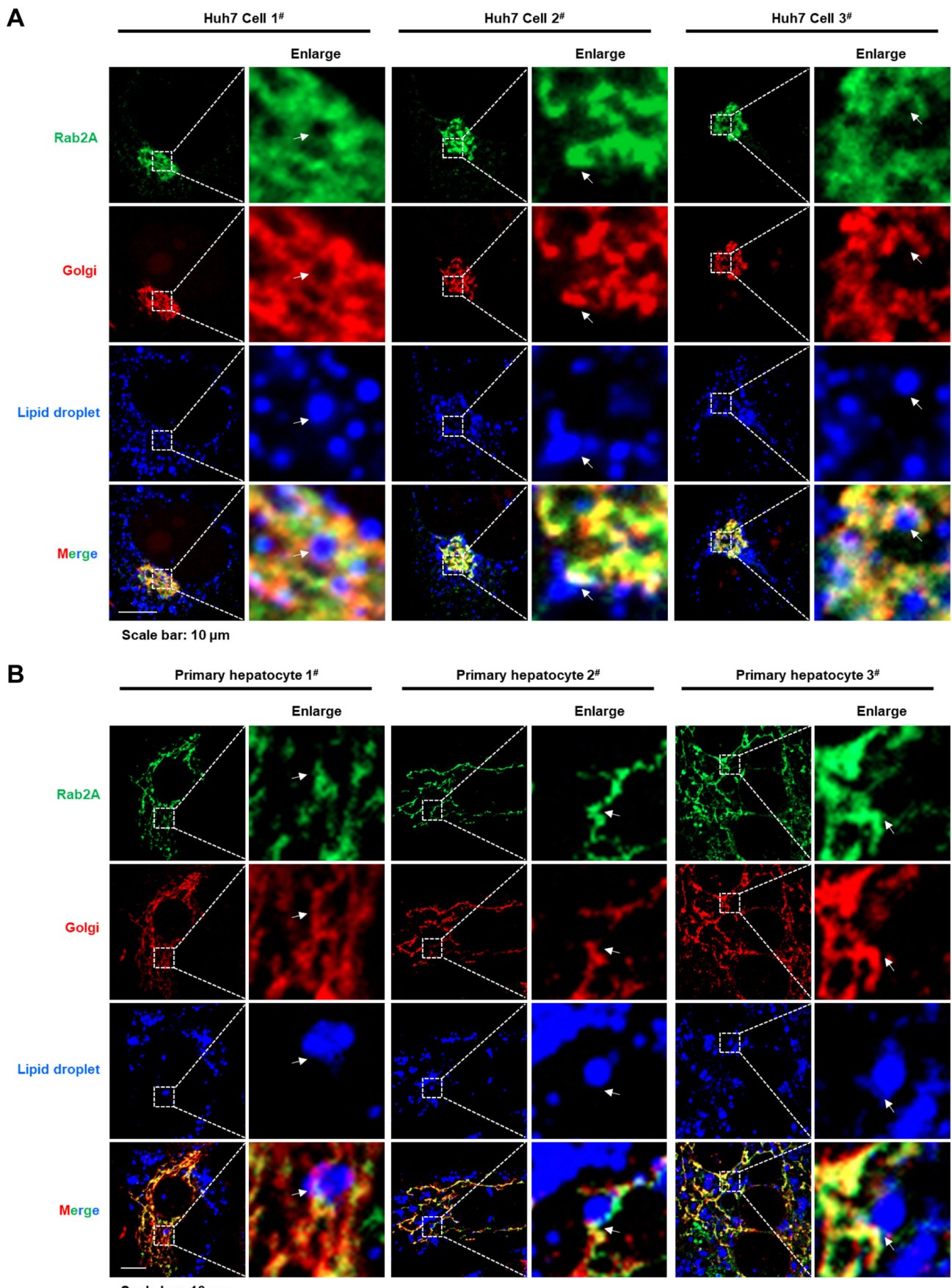

◄ **Figure EV2. Golgi-localized Ras-related protein Rab-2A (Rab2A) contacts with LDs.**

(A, B) Immunofluorescence analysis elucidated Rab2A-mediated organelle contacts between Golgi and LDs in Huh7 cells (A) and primary hepatocytes (B) with overexpression of EGFP-Rab2A plasmids. Golgi was labeled using a primary antibody targeting GM130, while LDs were stained with Bodipy (The white arrow indicates different types of contact points).

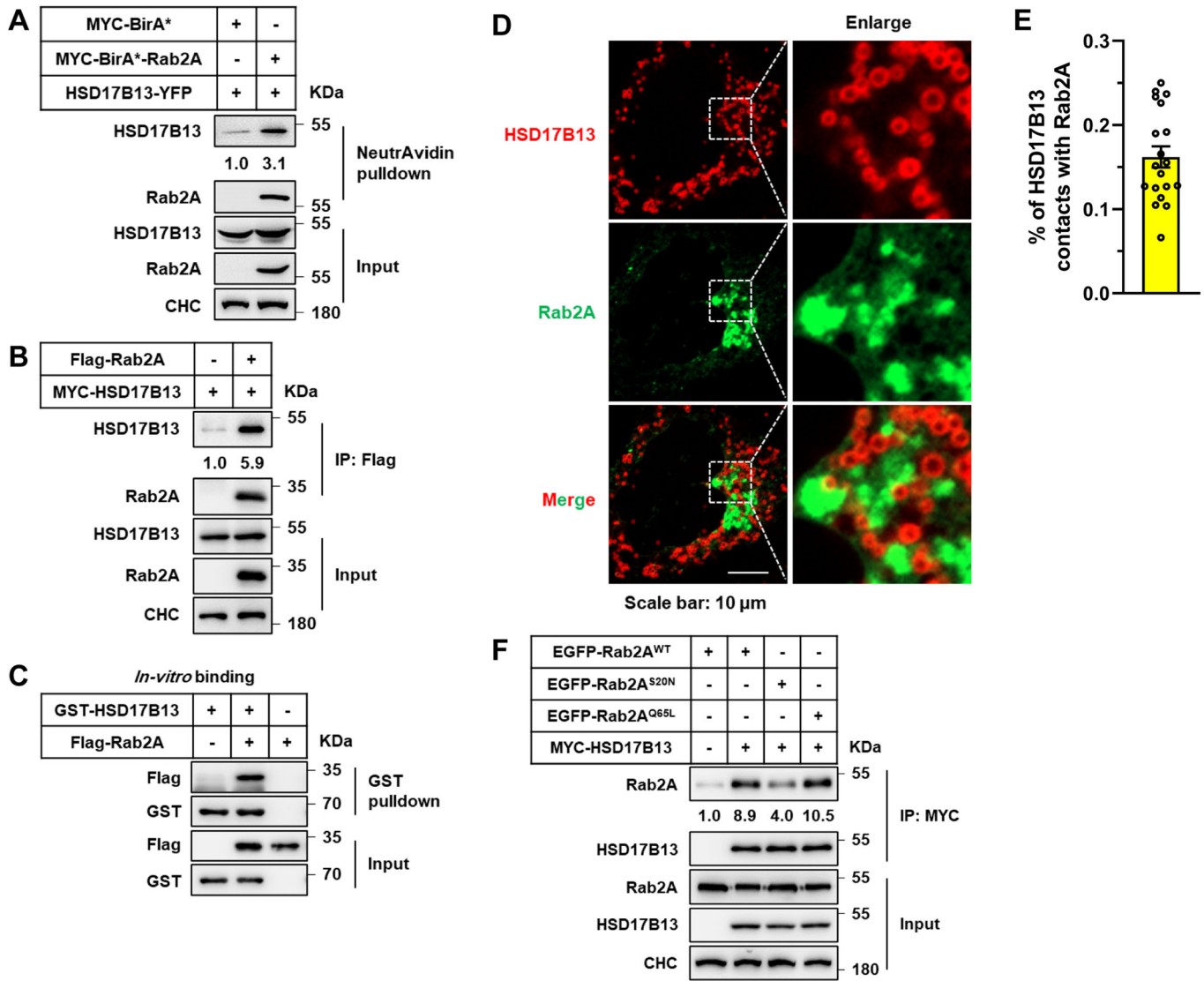

**Figure EV3. Confirming the binding efficiency between Rab2A and 17-beta-hydroxysteroid dehydrogenase 13 (HSD17B13) using exogenously expressed Rab2A and HSD17B13.**

(A) HEK293T cells were transfected with the BirA*-Rab2A and HSD17B13 plasmids. Rab2A-HSD17B13 binding was evaluated via NeutrAvidin beads pulldown, followed by visualization using Western blotting. (B) HEK293T cells were transfected with Flag-Rab2A and MYC-HSD17B13 plasmids, followed by a Flag-affinity beads pulldown assay to confirm Rab2A and HSD17B13 binding. (C) In vitro binding assay was conducted using purified GST-HSD17B13 and Flag-Rab2A proteins. (D, E) Huh7 cells underwent transfection with HSD17B13-mCherry and EGFP-Rab2A plasmids. Immunofluorescence was employed to observe the binding efficiency between Rab2A and HSD17B13. Representative images are shown (D), and the percentage of HSD17B13 contacting with Rab2A was quantified (E) ($n = 19$ cells). (F) The regulatory effect of Rab2A activity on the binding between Rab2A and HSD17B13 was examined in HEK293T cells by transfecting with different types of Rab2A plasmids, followed by MYC-affinity beads pulldown assay. Data information: Data in (E) are presented as mean ± SEM. Circles in (E) correspond to individual Huh7 cell. Source data are available online for this figure.

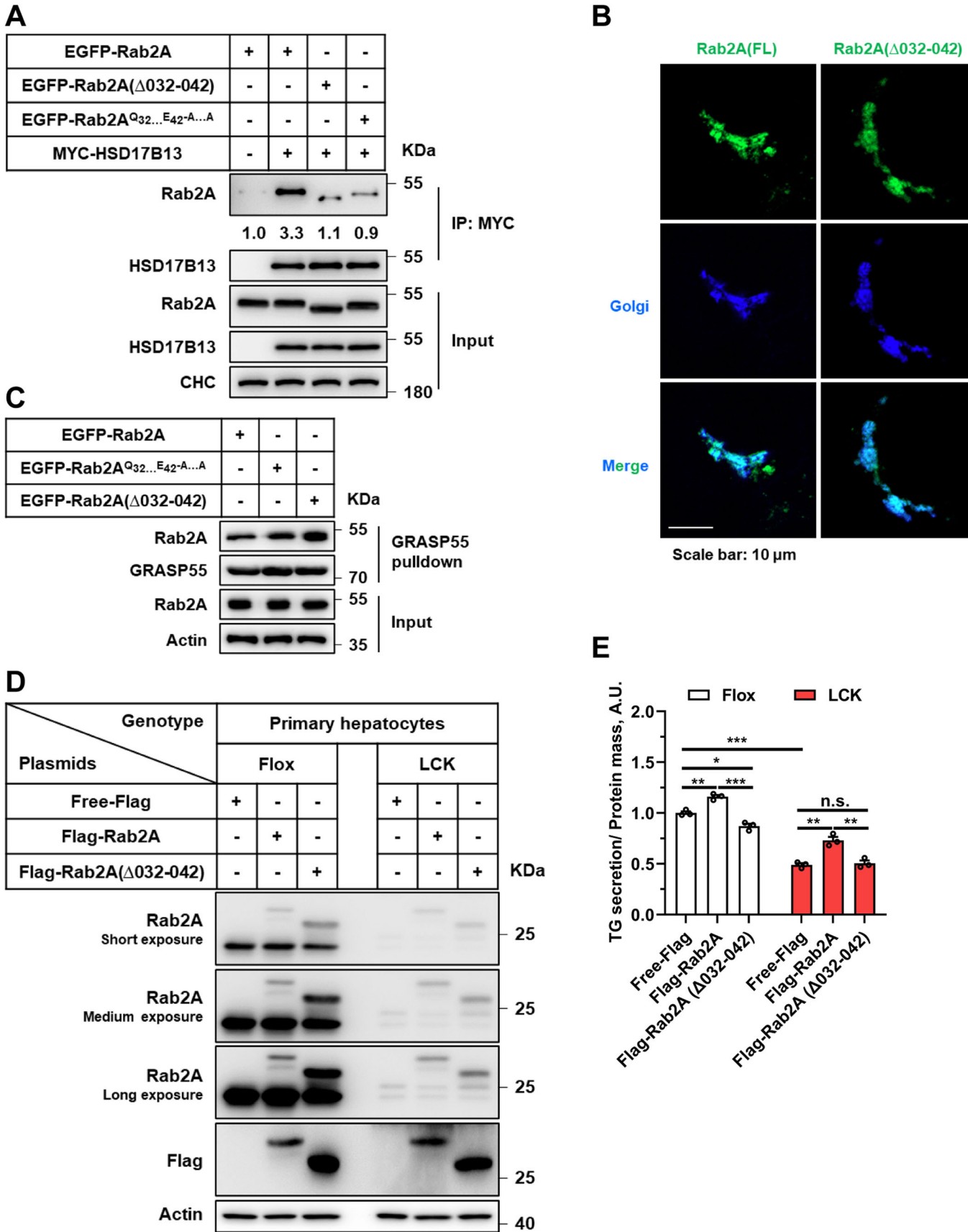

Scale bar: 10 μm

**Figure EV4.  Blocking the binding between Rab2A and HSD17B13 using a Rab2A mutant protein suppresses very-low-density lipoprotein (VLDL) secretion in primary hepatocytes.**

(A–E) Examination of the effects of Rab2A and HSD17B13 binding on TG secretion involving multiple steps. First, vital amino acids for complex binding in Rab2A were identified via Western blotting in HEK293T cells after transfection with relative plasmids (A). Second, the localization of mutant Rab2A protein was assessed using immunofluorescence in Huh7 cells following transfection with EGFP-Rab2A plasmids, with Golgi labeling accomplished using RCAS1-BFP (B). Third, the activity of both wild-type and mutant Rab2A proteins was detected using a GST-GRASP55 pulldown assay to assess influence of specific mutations on Rab2A's activity (C). Finally, the impact of deficient complex binding on TG secretion was evaluated in Flox and LCK primary hepatocytes after protein overexpression with the AAV-virus system (D). The medium was collected for testing, with values in Flox samples with Free-Flag overexpression normalized to 1 (E) (Flox hepatocytes (Free-Flag vs. Flag-Rab2A), $P = 0.0025$; Flox hepatocytes (Free-Flag vs. Flag-Rab2A (Δ032-042)), $P = 0.0101$; Flox hepatocytes (Flag-Rab2A vs. Flag-Rab2A (Δ032-042)), $P = 0.0005$; Free-Flag (Flox vs. LCK), $P < 0.0001$; LCK hepatocytes (Free-Flag vs. Flag-Rab2A), $P = 0.0030$; LCK hepatocytes (Flag-Rab2A vs. Flag-Rab2A (Δ032-042)), $P = 0.0063$). Data information: Data in (E) are presented as mean ± SEM. Circles in (E) correspond to individual test. $P$ values in (E) were determined using unpaired two-tailed Student's $t$-test. n.s. indicates no significant difference ($P > 0.05$), * indicates $P < 0.05$; ** indicates $P < 0.01$; *** indicates $P < 0.001$.

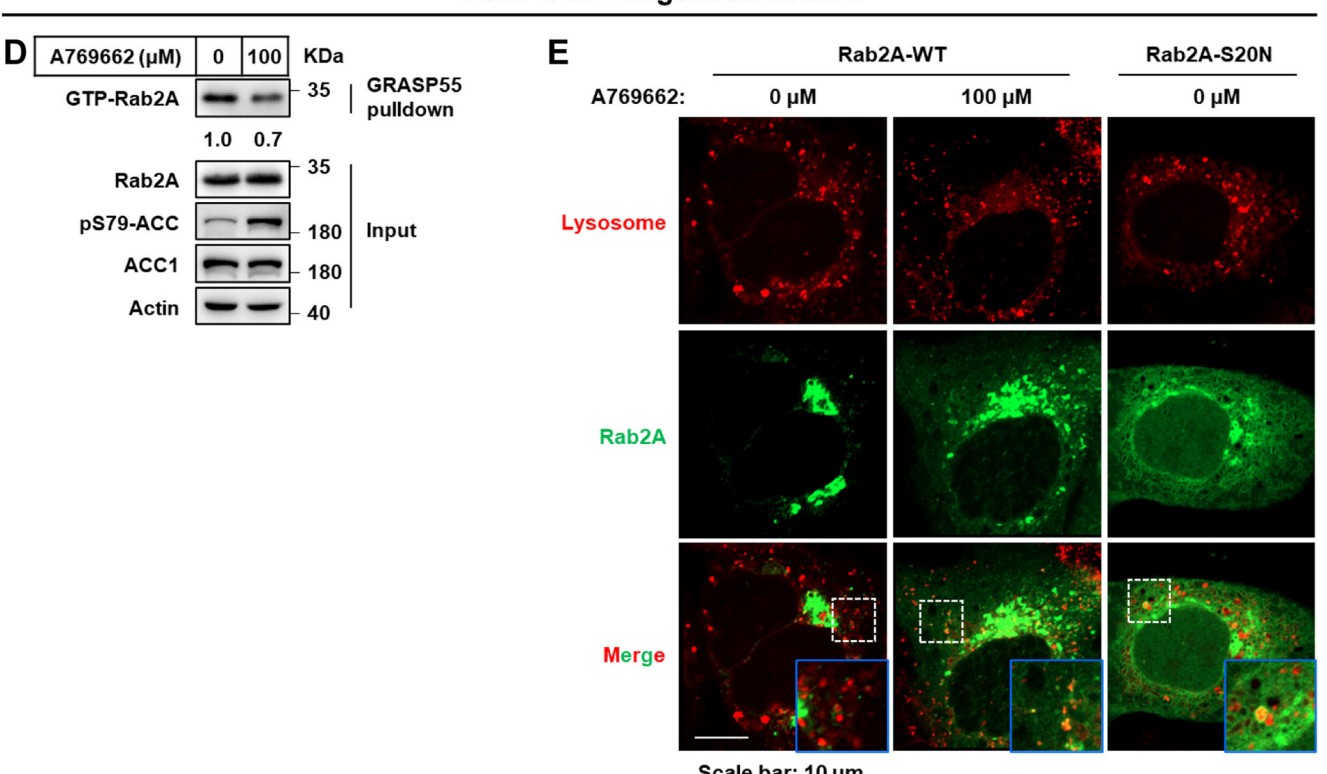

## Huh7 Cell-Endogenous Rab2A

## HEK293T Cell-Exogenous Rab2A

## Huh7 Cell-Exogenous Rab2A

Scale bar: 10 μm

◀  **Figure EV5.   A769662 stimulation attenuates Rab2A activity, Golgi-localization and Rab2A-HSD17B13 binding.**

(A–C) Huh7 cells were stimulated with A769662 (100 μM), an AMPK agonist, for 4 h in DMEM medium without fetal bovine serum (FBS). Rab2A activity was evaluated using the GRASP55 pulldown assay (A). Its subcellular localization was assessed with immunofluorescence (B). Endogenous Rab2A was stained with a primary antibody, and the Golgi apparatus was labeled with a primary antibody against GM130. The dispersion of Rab2A signaling was quantified via Li's ICQ value (Li's colocalization value) in Image J ($n = 44$ vs. 44 cells), with white arrows indicating potential signaling away from the Golgi (C) ($P < 0.0001$). (D, E) Huh7 cells, transfected with exogenously expressing Rab2A, were stimulated with A769662 (100 μM) for 4 h in DMEM medium without FBS. The activity of Rab2A was assessed using a GRASP55 pulldown assay and analyzed by Western blotting (D). The subcellular localization of Rab2A was evaluated with immunofluorescence, with lysosomes stained using LysoTracker (E). (F, G) HEK293T cells were transfected with exogenously expressing Rab2A plasmids, with or without co-transfection of HSD17B13 plasmids, and subsequently stimulated with A769662 (100 μM) for 4 h in DMEM medium without FBS. Rab2A activity was evaluated using the GRASP55 pulldown assay (F). The binding efficiency of Rab2A-HSD17B13 complex was assessed by an EGFP-affinity beads pulldown assay (G). Data information: Data in (C) are presented as mean ± SEM. Circles in (C) correspond to individual cell. $P$ value in (C) was determined using unpaired two-tailed Student's $t$-test. *** indicates $P < 0.001$.

                                                      