## [Peer Review File · The EMBO Journal]

Rab2A-mediated Golgi-Lipid droplet interactions support very-low-density lipoprotein secretion in hepatocytes.

Min Xu, Zi-yue Chen, Yang Li, Yue Li, Ge Guo, Rong-zheng Dai, Na Ni, Jing Tao, Hongyu Wang, Qiaoli Chen, Hua Wang, Hong Zhou, Yi-ning Yang, Shuai Chen, and Liang Chen

Corresponding author(s): Liang Chen (liang-chen@ahmu.edu.cn), Hong Zhou (hzhou@ahmu.edu.cn), Yi-ning Yang (yangyn5126@xjrmmy.com), Shuai Chen (chenshuai@nju.edu.cn)

Review Timeline:

Submission Date:	1st May 24
Editorial Decision:	4th Jun 24
Revision Received:	25th Aug 24
Editorial Decision:	27th Sep 24
Revision Received:	8th Oct 24
Editorial Decision:	14th Oct 24
Revision Received:	15th Oct 24
Accepted:	16th Oct 24

Editor: William Teale

Transaction Report:

Dear Dr. Chen,

Thank you again for the submission of your manuscript entitled "Golgi-Lipid droplet contact supports very-low-density lipoprotein secretion in hepatocytes" and for your patience during the review process. We have now received the reports from the referees, which I copy below.

As you can see from their comments, all referees found your manuscript interesting and timely. That said, all of them point out that the data you present on the Rab2A-HSD17B13 interaction, and the rationale for studying this pairing, will have to be revisited. I would like you carefully to consider whether you judge the reviewers' requests to be addressable in a time frame of six months (except for in vivo complementation).

Should this be feasible, we could consider a revised manuscript. I should add that it is The EMBO Journal policy to allow only a single major round of revision and that it is therefore important to resolve the main concerns at this stage. I recommend we take some time for a Zoom call to go over the referees' reports next week; please advise on your availability. Please, follow the instructions below when preparing your manuscript for resubmission.

I would also like to point out that as a matter of policy, competing manuscripts published during this period will not be taken into consideration in our assessment of the novelty presented by your study ("scooping" protection). We have extended this 'scooping protection policy' beyond the usual 3 month revision timeline to cover the period required for a full revision to address the essential experimental issues. Please contact me if you see a paper with related content published elsewhere to discuss the appropriate course of action.

Again, please contact me at any time during revision if you need any help or have further questions.

Thank you very much again for the opportunity to consider your work for publication. I look forward to your revision.

Best regards,

William

William Teale, Ph.D.
Editor
The EMBO Journal

When submitting your revised manuscript, please carefully review the instructions below and include the following items:

- 1) a .docx formatted version of the manuscript text (including legends for main figures, EV figures and tables). Please make sure that the changes are highlighted to be clearly visible.
- 2) individual production quality figure files as .eps, .tif, .jpg (one file per figure).
- 3) a .docx formatted letter INCLUDING the reviewers' reports and your detailed point-by-point response to their comments. As part of the EMBO Press transparent editorial process, the point-by-point response is part of the Review Process File (RPF), which will be published alongside your paper.
- 4) a complete author checklist, which you can download from our author guidelines ([https://wol-prod-cdn.literatumonline.com/pb-assets/embo-site/Author Checklist%20-%20EMBO%20J-1561436015657.xlsx](https://wol-prod-cdn.literatumonline.com/pb-assets/embo-site/Author%20Checklist%20-%20EMBO%20J-1561436015657.xlsx)). Please insert information in the checklist that is also reflected in the manuscript. The completed author checklist will also be part of the RPF.
- 5) Please note that all corresponding authors are required to supply an ORCID ID for their name upon submission of a revised manuscript.
- 6) We require a 'Data Availability' section after the Materials and Methods. Before submitting your revision, primary datasets produced in this study need to be deposited in an appropriate public database, and the accession numbers and database listed under 'Data Availability'. Please remember to provide a reviewer password if the datasets are not yet public (see <https://www.embopress.org/page/journal/14602075/authorguide#datadeposition>). If no data deposition in external databases is

needed for this paper, please then state in this section: This study includes no data deposited in external repositories. Note that the Data Availability Section is restricted to new primary data that are part of this study.

Note - All links should resolve to a page where the data can be accessed.

8) For data quantification: please specify the name of the statistical test used to generate error bars and P values, the number (n) of independent experiments (specify technical or biological replicates) underlying each data point and the test used to calculate p-values in each figure legend. The figure legends should contain a basic description of n, P and the test applied. Graphs must include a description of the bars and the error bars (s.d., s.e.m.).

9) We would also encourage you to include the source data for figure panels that show essential data. Numerical data can be provided as individual .xls or .csv files (including a tab describing the data). For 'blots' or microscopy, uncropped images should be submitted (using a zip archive or a single pdf per main figure if multiple images need to be supplied for one panel). Additional information on source data and instruction on how to label the files are available at .

10) We replaced Supplementary Information with Expanded View (EV) Figures and Tables that are collapsible/expandable online (see examples in <https://www.embopress.org/doi/10.15252/embj.201695874>). A maximum of 5 EV Figures can be typeset. EV Figures should be cited as 'Figure EV1, Figure EV2" etc. in the text and their respective legends should be included in the main text after the legends of regular figures.

12) Our journal encourages inclusion of *data citations in the reference list* to directly cite datasets that were re-used and obtained from public databases. Data citations in the article text are distinct from normal bibliographical citations and should directly link to the database records from which the data can be accessed. In the main text, data citations are formatted as follows: "Data ref: Smith et al, 2001" or "Data ref: NCBI Sequence Read Archive PRJNA342805, 2017". In the Reference list, data citations must be labeled with "[DATASET]". A data reference must provide the database name, accession number/identifiers and a resolvable link to the landing page from which the data can be accessed at the end of the reference. Further instructions are available at .

- a point-by-point response to the referees' comments, with a detailed description of the changes made (as a word file).
- a word file of the manuscript text
- individual production quality figure files (one file per figure)
- a complete author checklist, which you can download from our author guidelines (<https://www.embopress.org/page/journal/14602075/authorguide>).
- Expanded View files (replacing Supplementary Information)

The revision must be submitted online within 90 days; please click on the link below to submit the revision online before 2nd Sep 2024.

Referee #1:

In "Golgi-Lipid droplet contact supports very-low-density lipoprotein secretion in hepatocytes", Xu and colleagues propose a role for an interaction between Rab2A and HSD17B13 at lipid droplet-Golgi contact sites in the lipidation of lipoproteins in the Golgi for secretion and the role of SNARE proteins in lipid transfer at these sites. The authors began their investigation with the observation that Rab2A knockout in murine hepatocytes decreases serum triglyceride (TG) and total cholesterol (TC) levels without modifying the apolipoproteins levels in serum compared to control, while accumulating these lipid species and APOB within hepatocytes. They then suggest that Rab2A produces this effect due to changes in VLDL2 lipidation in the Golgi. They demonstrate Rab2A localization to the Golgi in an activity-dependent manner. Next, building on an observation of Rab2A at lipid droplet (LD) contacts with the Golgi, they demonstrate associations between LD and Golgi reduce with the loss of Rab2A using an in vitro reconstitution system. They then identified HSD17B13 as an LD localized binding partner of Rab2A. Like Rab2A, Xu et al demonstrate an impact of HSD17B13 on serum TG and TC levels, although the effect may be compounded by a role of HSD17B13 in APOB protein secretion. To understand how lipids transfer between LD and Golgi, they show an interaction between two SNAREs, LD localized VAMP-4 and Golgi localized GS28. Finally, they demonstrate that fasting disrupts this interaction, concluding that Rab2A activity may mediate this process.

This manuscript describes a novel function for LD-Golgi contact sites and provides a useful insight into the mechanism of VLDL lipidation. We also appreciated the characterization of the interaction between Rab2A and HSD17B13 and the inclusion of a potential mechanism for lipid transfer in the form of the SNARE complex formation in the final data figure. We also appreciate the use of in vivo models to provide a physiological context for the mechanistic data obtained through cell culture systems.

Overall, we found the manuscript strong. We believe that the manuscript would benefit from increased evidence of the role of Rab2A at LD-Golgi contact sites in situ in cells rather than only through an in vitro reconstitution. If the authors can address this major comment and the minor comments below, we recommend publication.

Major comment:

While the authors do provide a representative micrograph image demonstrating Rab2A proximity to LD-Golgi contact sites in Figure 4, we would like to see more quantification of this interaction. While the manuscript benefited from demonstrating a role for Rab2A in facilitating these contacts using in vitro reconstitution, we think that the manuscript would be strengthened significantly through quantifying LD-Golgi contacts and Rab2A proximity within their system at baseline and in a fasted and/or AMPK agonist-treated condition. We would also like the authors to test whether the variation of these contacts is dependent on Rab2A or HSD17B13 within a cell culture or animal model system, e.g., by knocking down Rab2A and/or HSD17B13. While the evidence for an interaction between these two proteins within the paper is strong, more evidence for the specific role of the LD in particular for this process could be provided.

Minor comments:

1. In Figure 2, no n is provided for panels D, E, F, G, H, I and J. If D-F and G- H are from a single replicate, replication should be performed. For I and J, the number of animals and total number of particles in each category should be provided.
2. In Figure 3G, the degree of colocalization, while evident from the representative image, would benefit from quantification. The authors also claim the Rab2A relocates from the Golgi to the cytoplasm, and, while the change in localization is evident, the localization may be to other organelles (e.g., ER) based on the pattern within the representative image. Within the text, the authors should either address this potential localization or perhaps simply reference a loss of Golgi localization.
3. All the in vivo experiments were performed with male animals. The authors should mention this as a caveat and consider including both sexes in future studies since is now considered best practice.
4. Two very recent papers came out about LD-Golgi contacts that the authors should cite:

- Du Y, Hu X, Chang W, Deng L, Ji WK, Xiong J. A Possible Role of VPS13B in the Formation of Golgi-Lipid Droplet Contacts Associating with the ER. *Contact* (Thousand Oaks). 2023 Sep 11;6:25152564231195718. doi: 10.1177/25152564231195718. PMID: 38090145; PMCID: PMC10714374.

- Sherman DJ, Liu L, Mamrosh JL, Xie J, Ferbas J, Lomenick B, Ladinsky MS, Verma R, Rulifson IC, Deshaies RJ. The fatty liver disease-causing protein PNPLA3-I148M alters lipid droplet-Golgi dynamics. *Proc Natl Acad Sci U S A*. 2024 Apr 30;121(18):e2318619121. doi: 10.1073/pnas.2318619121. Epub 2024 Apr 24. PMID: 38657050; PMCID: PMC11067037.
5. Fig. 5G minor typo "lipid drolet" instead of "lipid droplet".

Referee #2:

Xu et al. have presented a compelling story exploring the underlying cell biology of VLDL synthesis and secretion. Using a Rab2A KO mouse, they observe a decrease in proteolipid abundance, which they determine is coupled with a decrease in the Golgi apparatus. They confirm the known localisation of Rab2A to the Golgi apparatus and show that Rab2A KO affects the tethering of the Golgi and lipid droplets. They identify, a novel Rab2A effector named HSD17B13 a lipid oxidizing enzyme, which they propose acts as the link between the Golgi and lipid droplets through an interaction with Rab2A. Finally, they implicate a SNARE complex, which includes VAMP4 as part of this contact site. The manuscript and findings are compelling and exciting, and the experiments are presented well. For the most part, the manuscript is clear. I have several concerns about some key experiment conceptual mechanisms, controls, repeats, and statistics. In addition, some of the interpretations lacked clarity for me. I have detailed my concerns below.

In the KO a decrease in VLDL triglycerides is detected, the authors interpret this as a biosynthetic defect, however, in the light of the increase in the abundance of APOB100 and APOB48, another interpretation would be either a systemic defect in turnover or a defect in lipoprotein uptake or metabolism. The authors use tyloxapol, which inhibits lipoprotein lipase, however the interpretation of this is complicated as tyloxapol is a surfactant so the effects on lipid synthesis and metabolism may also be nebulous. How do the authors resolve these possibilities?

For several absolutely crucial experiments, the authors purify the Golgi apparatus. However, these miss essential controls, including interrogating the Golgi membrane for other organelle markers, including the ER, endosomes, cytosol contamination, and lipid droplet proteins. In addition, there is a lack of clarity on the number of experimental repeats for some of these experiments. I know it is fastidious, however many of the presented experiments (eg and not limited to- Rab IPs, organelle fractionations, the measurement of the VLDL particules) are notoriously challenging, and having proper experimental repeats and statistical analysis for these and other key experiments is important. I know the authors say, "Each experiment was independently replicated at least thrice, yielding consistent results" - however, for key experiments, this is not evidenced.

Identifying a novel effector with a potential function is a key result. However, 1) the localisation between HSD17B13 and Rab2A is very weak. One would expect them to be, at least partly, in the same place if this was robust, and 2) the authors should test HSD17B13 KO in their tethering assay in 4E in addition to the PLA. The PLA is very weak, and a close examination of the images shows a random pattern, with one highlighted spot at a contact site. I do not see repeats or quantification, and this alone makes me very concerned about the intention of the interpretation here and throughout.

The authors implicate SNAREs in their proposed mechanism. I find this poorly justified and, given the novelty of this proposed mechanism, weakly evidenced. In a way, this alone would be significant enough for a publication, but in its current form, there are many other interpretations of the data. SNAREs are thought to mediate lipid droplet fusion, so it is already known that they localise there, and the mechanistic insight into the final model is, to my mind, not present.

Minor

I think showing a full gel for the Rab2A KO is crucial, as a single exon is deleted. A truncation protein may be made, which would not affect any interpretations but is important for future experimenters.

"Intriguingly, the Golgi localization of Rab2A appears largely contingent upon its activation state; GTP-bound Rab2A, signifying active status, preferentially localizes to the Golgi, as shown by overexpressing a constitutively active Rab2A mutant (Q65L)." - I am not sure why this is intriguing, as it is exactly how one would expect a Rab to function. Activated rabs are recruited to membranes.

Rab2A's localisation to the Golgi is not novel; however, Rab2A has been reported to localise to the ERGIC. The localisation quantification in this figure is substandard, and proper repeats and quantification should be undertaken.

"These experiments unveiled that fasting markedly dampened Rab2A activity, subsequently reducing its association with Golgi (Fig. 3H, I)" - Again, the controls showing the purification of the compartment are important. In addition, this assay does not directly show Rab2A activity and should not be reported as such.

"Rab2A may act as a tethering molecule" - I understand the authors' intention here. However, this is perhaps worth rewording,

as Rab2A will recruit a number of effectors, one of which may be involved in tethering.

Perhaps I am wrong; however, I suspect the authors have used ChatGPT to assist with the writing of this manuscript. I have no problem with this in principle as a reviewer, as on the whole the manuscript is clear, however there are number of sections where the writing needs to be modified. Particularly in the figure legends where phrases such as "Extensive assays were conducted", "The graphical abstract delineates", "schematic depiction elucidates", and "The binding efficiency was meticulously assessed step by step using Flag affinity or MYC-affinity beads pulldown assays."

Referee #3:

The manuscript titled "Golgi-lipid droplet contact supports very-low-density lipoprotein secretion in hepatocytes" presents a study on lipid delivery from lipid droplets to the Golgi apparatus, modulated by Rab2A. This research aims to elucidate the molecular mechanisms underlying VLDL secretion in hepatocytes. The initial observations show that Rab2A depletion results in decreased serum triglycerides (TG) and total cholesterol (TC) levels and increased serum APOB levels. Further investigation reveals that these decreased lipid levels are due to reduced VLDL release rather than uptake (tyloxapol treatment). Intracellularly, overexpressed Rab2A-GFP localizes to the Golgi, and its deactivation (A769662 treatment) leads to Rab2A dissociating from the Golgi. Using immunofluorescence, protein ligation assays, and immunoprecipitation, the authors tested the hypothesis that Rab2A interacts with lipid droplets via activated Rab2A's interaction with HSD17B13. The paper provides evidence that Rab2A-liverKO mice have impaired VLDL release, potentially due to disrupted lipid delivery from lipid droplets to the Golgi, where APOB lipidation occurs. However, several critical issues compromise the clarity and impact of the findings.

First, the rationale for selecting Rab2A as a central player in lipid droplet-Golgi interactions is inadequately justified. The manuscript lacks a comprehensive introduction to Rab2A's known functions and its significance in lipid metabolism. The study's foundation appears weak without a clear explanation of why Rab2A was chosen for investigation. The authors should provide a thorough background on Rabs and SNARE proteins, detailing their roles in intracellular trafficking and lipid metabolism. This would help readers understand the relevance of Rab2A in the context of VLDL secretion.

Furthermore, the study relies heavily on overexpression systems, which can lead to artefactual results that do not accurately represent physiological conditions. Overexpression of proteins, such as Rab2A and SNAREs, can result in non-physiological interactions and localization, potentially skewing the study's conclusions. The authors should address the limitations of using overexpression models and provide data from endogenous protein levels to strengthen their claims.

The immunofluorescence images are difficult to interpret, and the rationale behind the experiments is inadequately explained. The use of different cell lines in different experiments adds confusion, and the data does not consistently support the claims or the manuscript title. The interaction has not been demonstrated for endogenously expressed proteins, nor is there evidence for the endogenous STX5-GS28-GS15-VAMP4 SNARE complex or endogenous Rab2A or VAMP4 interaction with lipid droplets. It is also puzzling how transmembrane VAMP4 can be associated with lipid droplets.

In summary, despite extensive data, the manuscript fails to provide significant mechanistic insights into the Golgi's role in lipid droplet biogenesis.

Specific Comments:

Title: The manuscript title is too broad. A more specific title reflecting the findings would be better.

Rationale for Rab2A: The authors should elaborate on the choice of Rab2A as an interacting partner of lipid droplets, as this is a crucial aspect of their research. The absence of a description of Rabs and SNAREs is a notable gap. The rationale for selecting Rab2A should be adequately explained in the Introduction.

Knockout Models: No evidence is provided that the phenotype in knockout models could be rescued by reintroducing Rab2A.

Figure 1: The discrepancy in units used in Figure 1A, 1B (mg/dl) and Figure 1G, H (mmol/ul) for TG and TC needs clarification. Additionally, the rescue experiments should be more pronounced and better explained.

Figure 3: The authors should clarify the novelty of their finding that GTP-bound Rab2A is primarily located in the Golgi, as this has been previously demonstrated (PMID: 25377857, PMID: 30957628).

Golgi Morphology: In Figure 3E and 3G, the authors concluded that Golgi morphology is not affected in LCK. However, the electron microscopy images suggest otherwise. Quantification of the Golgi morphology should be provided.

AMPK Agonist: The use of A769662 to reduce Rab2A activity and its effects on Golgi morphology should be better quantified and explained. The Rab2A mislocalization due to overexpression needs further clarification.

Figures 4A and 4B: The authors should provide separate graphs for Golgi and ER TC and TG levels. The error bars for ER are challenging to see, and quantification for Figure 4C is missing.

Figure 4D: Only one cell image showing colocalization is insufficient to demonstrate the interaction between Rab2A and lipid droplets. Colocalization studies and quantification are necessary.

Figure 4E and 4F: The lipid droplet pulldown experiment lacks detailed methodology and evidence. The experiment does not adequately explain the Golgi-LD interface.

Figure 5: The overexpression of proteins in the experiments raises concerns about the findings. Colocalization statistics are missing, and the rationale for using RCAS1-BFP for Golgi labeling is unclear.

Golgi Fragmentation: The fragmented Golgi in Figure 5F and EV3 needs further investigation and explanation.

Figure 6B: The explanation of PLA blobs is inadequate. The differences between treated and untreated cells are difficult to discern from the IF images.

VLDL Secretion: The authors claim that the Rab2A-HSD17B13 complex mediates VLDL secretion, but they provide no direct data supporting VLDL secretion.

SNARE Proteins: The study on SNARE proteins lacks proper citations and rationale. Overexpression studies do not convincingly demonstrate SNARE interactions.

Figure 7B: The image quality is poor, and it is challenging to identify GS28 and VAMP4. The SNARE interaction claims require more robust evidence.

GS28 Distribution: The increased Golgi localization of GS28 in LCK mice and its implications for SNARE complex formation need clarification. The role of other SNARE partners should be discussed.

Novel SNARE Complex: The claim of identifying a novel SNARE complex (Syntaxin-5/GS28/GS15/VAMP-4) is unsubstantiated. More robust data is needed.

Model in Figure 8: The model explaining Rab2A's role in Golgi-LD contacts and VLDL secretion is poorly detailed. The discussion on VLDL1 and VLDL2 is missing.

Discussion: The discussion section needs significant improvement for clarity and coherence.

Dear Editor William Teale,

We sincerely appreciate the constructive comments and suggestions provided by you and the reviewers, which have greatly helped us to improve the quality of our manuscript. In response, we have obtained amounts of new data to strengthen our study and conclusions. Specifically, we have confirmed the interaction between endogenously expressed Rab2A and HSD17B13, and demonstrated their roles in mediating Golgi-LD dynamic contacts. In addition, in alignment with the reviewers' suggestions, we have carefully reconsidered the manuscript's logical flow and removed the SNARE complex studies. Below, we provide detailed point-by-point responses to the reviewers' comments.

Referee #1:

In "Golgi-Lipid droplet contact supports very-low-density lipoprotein secretion in hepatocytes", Xu and colleagues propose a role for an interaction between Rab2A and HSD17B13 at lipid droplet-Golgi contact sites in the lipidation of lipoproteins in the Golgi for secretion and the role of SNARE proteins in lipid transfer at these sites. The authors began their investigation with the observation that Rab2A knockout in murine hepatocytes decreases serum triglyceride (TG) and total cholesterol (TC) levels without modifying the apolipoproteins levels in serum compared to control, while accumulating these lipid species and APOB within hepatocytes. They then suggest that Rab2A produces this effect due to changes in VLDL2 lipidation in the Golgi. They demonstrate Rab2A localization to the Golgi in an activity-dependent manner. Next, building on an observation of Rab2A at lipid droplet (LD) contacts with the Golgi, they demonstrate associations between LD and Golgi reduce with the loss of Rab2A using an in vitro reconstitution system. They then identified HSD17B13 as an LD localized binding partner of Rab2A. Like Rab2A, Xu et al demonstrate an impact of HSD17B13 on serum TG and TC levels, although the effect may be compounded by a role of HSD17B13 in APOB protein secretion. To understand how lipids transfer between LD and Golgi, they show an interaction between two SNAREs, LD localized VAMP-4 and Golgi localized GS28. Finally, they demonstrate that fasting disrupts

this interaction, concluding that Rab2A activity may mediate this process.

This manuscript describes a novel function for LD-Golgi contact sites and provides a useful insight into the mechanism of VLDL lipidation. We also appreciated the characterization of the interaction between Rab2A and HSD17B13 and the inclusion of a potential mechanism for lipid transfer in the form of the SNARE complex formation in the final data figure. We also appreciate the use of in vivo models to provide a physiological context for the mechanistic data obtained through cell culture systems.

Overall, we found the manuscript strong. We believe that the manuscript would benefit from increased evidence of the role of Rab2A at LD-Golgi contact sites in situ in cells rather than only through an in vitro reconstitution. If the authors can address this major comment and the minor comments below, we recommend publication.

Major comment:

While the authors do provide a representative micrograph image demonstrating Rab2A proximity to LD-Golgi contact sites in Figure 4, we would like to see more quantification of this interaction. While the manuscript benefited from demonstrating a role for Rab2A in facilitating these contacts using in vitro reconstitution, we think that the manuscript would be strengthened significantly through quantifying LD-Golgi contacts and Rab2A proximity within their system at baseline and in a fasted and/or AMPK agonist-treated condition. We would also like the authors to test whether the variation of these contacts is dependent on Rab2A or HSD17B13 within a cell culture or animal model system, e.g., by knocking down Rab2A and/or HSD17B13. While the evidence for an interaction between these two proteins within the paper is strong, more evidence for the specific role of the LD in particular for this process could be provided.

Response: Thanks for your kind comments. As you know, our study initiated with a comprehensive analysis of Rab2A liver-specific knockout mice, focusing on its impact on VLDL lipidation and secretion. We then elucidated a mechanism in which Golgi-localized Rab2A interacts with HSD17B13, a lipid droplet-resident protein,

facilitating dynamic contacts between the Golgi apparatus and lipid droplets, ultimately contributing to VLDL₂ lipidation within the Golgi (Fig. 6).

To substantiate the role of the Rab2A-HSD17B13 complex in mediating Golgi-lipid droplet contacts, we confirmed the subcellular localization of endogenously expressed Rab2A on the Golgi (Fig. 3A, B). Furthermore, we demonstrated that Golgi-localized Rab2A established contacts with lipid droplets in both primary hepatocytes and Huh7 cells (Fig. EV2). Rab2A deficiency led to a reduction in the number and level of contacts between the Golgi and lipid droplets (Fig. 3H-J).

Additionally, we showed that lipid droplet-localized HSD17B13 bound with Rab2A (Fig. 4A-C; EV3), and that inhibition of HSD17B13 in primary hepatocytes decreased the contacts between the Golgi and lipid droplets (Fig. 4E, F, I). Moreover, blocking HSD17B13 or the interaction between Rab2A and HSD17B13 also resulted in reduced VLDL secretion (Fig. 4G, J, K; EV4).

Finally, our results indicated that AMPK activation, whether through fasting in the liver or A769662 stimulation in the primary hepatocytes and Huh7 cells, diminished Rab2A activity and its Golgi residency (Fig. 5A-C; EV5). A769662 incubation further reduced the contacts between the Golgi and lipid droplets (Fig. 5D-F), as well as VLDL secretion (Fig. 5G). Collectively, these data provide strong evidence that the Rab2A-HSD17B13 complex plays a crucial role in regulating Golgi-lipid droplet contacts, thereby influencing VLDL lipidation and secretion in hepatocytes.

Minor comments:

1. In Figure 2, no n is provided for panels D, E, F, G, H, I and J. If D-F and G- H are from a single replicate, replication should be performed. For I and J, the number of animals and total number of particles in each category should be provided.

Response: Thank you for this kind suggestion. Actually, the experiments in Figure 2D-2F and 2G-2H were replicated twice with similar patterns (1 vs. 1). Then, in Figure 2I-2J, the VLDL fraction in serum was initially collected and mixed from mice (3 vs. 3), then the diameter of VLDL particles was quantified over 200 particles. We have revised the description in the corresponding figure legends.

2. In Figure 3G, the degree of colocalization, while evident from the representative image, would benefit from quantification. The authors also claim the Rab2A relocalizes from the Golgi to the cytoplasm, and, while the change in localization is evident, the localization may be to other organelles (e.g., ER) based on the pattern within the representative image. Within the text, the authors should either address this potential localization or perhaps simply reference a loss of Golgi localization.

Response: Thanks for your suggestions, we repeated and quantified the assay, demonstrating that suppression of Rab2A activity with A769662 induced a dispersed distribution of Rab2A (Fig. EV5A-C). Subsequently, our studies further revealed the partial re-localization of Rab2A from Golgi to lysosome upon inactivation (Fig. EV5D, E), contributing to phagosome formation, a phenomenon also supported by previous findings (PMID:30957628, 28424218).

3. All the in vivo experiments were performed with male animals. The authors should mention this as a caveat and consider including both sexes in future studies since is now considered best practice.

Response: Thanks for your comments. In our study, we also detected the VLDL secretion rate with tyloxapol injection with female mice (Figure EV1A). However, we know it is not sufficient, and we will follow the standard, as you pointed out, in the future studies.

4. Two very recent papers came out about LD-Golgi contacts that the authors should cite:

- Du Y, Hu X, Chang W, Deng L, Ji WK, Xiong J. A Possible Role of VPS13B in the Formation of Golgi-Lipid Droplet Contacts Associating with the ER. *Contact* (Thousand Oaks). 2023 Sep 11;6:25152564231195718. doi: 10.1177/25152564231195718. PMID: 38090145; PMCID: PMC10714374.

- Sherman DJ, Liu L, Mamrosh JL, Xie J, Ferbas J, Lomenick B, Ladinsky MS, Verma R, Rulifson IC, Deshaies RJ. The fatty liver disease-causing protein

PNPLA3-I148M alters lipid droplet-Golgi dynamics. Proc Natl Acad Sci U S A. 2024 Apr 30;121(18):e2318619121. doi: 10.1073/pnas.2318619121. Epub 2024 Apr 24. PMID: 38657050; PMCID: PMC11067037.

Response: Thanks for your suggestions, we have cited the papers in our manuscript.

5. Fig. 5G minor typo "lipid drolet" instead of "lipid droplet".

Response: Thanks for your kind suggestions. We initially intended to repeat the PLA assay to address concerns about the image quality raised by other reviewers. Unfortunately, the delivery time for the necessary reagents (Duolink PLA kit, No. DUO92101, Sigma-Aldrich) exceeded the manuscript's revision deadline. As a result, we removed the PLA data and instead provided new data to demonstrate the binding between endogenous Rab2A and HSD17B13 (Figure 4A, C).

Referee #2:

Xu et al. have presented a compelling story exploring the underlying cell biology of VLDL synthesis and secretion. Using a Rab2A KO mouse, they observe a decrease in proteolipid abundance, which they determine is coupled with a decrease in the Golgi apparatus. They confirm the known localization of Rab2A to the Golgi apparatus and show that Rab2A KO affects the tethering of the Golgi and lipid droplets. They identify, a novel Rab2A effector named HSD17B13 a lipid oxidizing enzyme, which they propose acts as the link between the Golgi and lipid droplets through an interaction with Rab2A. Finally, they implicate a SNARE complex, which includes VAMP4 as part of this contact site. The manuscript and findings are compelling and exciting, and the experiments are presented well. For the most part, the manuscript is clear. I have several concerns about some key experiment conceptual mechanisms, controls, repeats, and statistics. In addition, some of the interpretations lacked clarity for me. I have detailed my concerns below.

In the KO a decrease in VLDL triglycerides is detected, the authors interpret this as a biosynthetic defect, however, in the light of the increase in the abundance of APOB100 and APOB48, another interpretation would be either a systemic defect in turnover or a defect in lipoprotein uptake or metabolism. The authors use tyloxapol, which inhibits lipoprotein lipase, however the interpretation of this is complicated as tyloxapol is a surfactant so the effects on lipid synthesis and metabolism may also be nebulous. How do the authors resolve these possibilities?

Response: Thank you for this comment. First of all, our story demonstrated that Rab2A knockout in the liver dramatically decreased serum lipids (Fig. 1), primarily due to the deficient VLDL secretion rather than impaired lipoprotein uptake (Fig. 2). To support this conclusion and address the limitations of a single assay, we initially evaluated lipoprotein uptake (Appendix Fig. S3B, C) and secretion rates (Fig. EV1C, D) in primary hepatocytes. We then performed lipid uptake assay (oil gavage) (Appendix Fig. S3A) and VLDL secretion assay (tyloxapol injection) (Fig. 2A, B; EV1A, B) in Flox and LCK mice. Additionally, LDLR knockout mice, which exhibit

severe hyperlipidemia due to deficient lipoprotein uptake, were used to assess lipoprotein uptake rates (Appendix Fig. S3D-H). Furthermore, Apo B inhibition to block VLDL secretion in Flox and LCK mice was conducted (Fig. EV1E-H). All these assays confirmed that Rab2A primarily influences VLDL secretion rather than lipoprotein uptake.

Subsequently, our extensive assays indicated that the attenuated VLDL lipidation levels in the Golgi and serum of LCK mice contributed to hypolipidemia (Figure 2D-J). VLDL is initially assembled in the ER, transported to the Golgi for further lipidation, and then secreted into the serum. Apo B protein, as the core of VLDL, is synthesized in the ER. Our studies showed that Rab2A deficiency led to the accumulation of Apo B in the liver, particularly in the Golgi apparatus (Fig. 3G; Appendix Fig. S2C, D), along with other relative apolipoproteins with similar expression levels in the liver (Appendix Fig. S2C, D). This potentially suggests that the reduced lipidation of VLDL is not due to the deficiencies in apolipoproteins. Interestingly, lipid levels were reduced in the Golgi but not in the ER (Fig. 3C-F), indicating that lipid depletion in the Golgi impairs VLDL₂ lipidation, thereby disrupting VLDL₁ formation and secretion. This disruption likely results in compensatory accumulation of Apo B, potentially due to increased protein stability

For several absolutely crucial experiments, the authors purify the Golgi apparatus. However, these miss essential controls, including interrogating the Golgi membrane for other organelle markers, including the ER, endosomes, cytosol contamination, and lipid droplet proteins. In addition, there is a lack of clarity on the number of experimental repeats for some of these experiments. I know it is fastidious, however many of the presented experiments (e.g. and not limited to- Rab IPs, organelle fractionations, the measurement of the VLDL particles) are notoriously challenging, and having proper experimental repeats and statistical analysis for these and other key experiments is important. I know the authors say, "Each experiment was independently replicated at least thrice, yielding consistent results" - however, for key experiments, this is not evidenced.

Response: Thanks for your comments. The methods described in our paper for the purification of Golgi apparatus are well-established and validated by multiple independent groups (PMID: 32103509, 22661308, 12960170). Here, we repeated the assay and detected markers for other organelles, including the ER, LD, cytosol and nucleus (Appendix Fig. S4). Additionally, the number of experimental repeats has been specified and revised in the corresponding figure legends.

Identifying a novel effector with a potential function is a key result. However, 1) the localization between HSD17B13 and Rab2A is very weak. One would expect them to be, at least partly, in the same place if this was robust, and 2) the authors should test HSD17B13 KO in their tethering assay in 4E in addition to the PLA. The PLA is very weak, and a close examination of the images shows a random pattern, with one highlighted spot at a contact site. I do not see repeats or quantification, and this alone makes me very concerned about the intention of the interpretation here and throughout.

Response: Thanks for your comments. Our extensive data strongly supports the interactions between Rab2A and HSD17B13, including endogenously expressed proteins (Fig. 4A; EV3; EV4). Additionally, we quantified the percentage of HSD17B13 in contact with Rab2A in Huh7 cells (Fig. EV3D, E). Importantly, our findings revealed partial tight contacts between endogenously expressed Rab2A and HSD17B13 in primary hepatocytes (Figure 4C).

We initially planned to repeat the PLA assay to address concerns about the image quality, as you suggested. However, due to the extended delivery time for the necessary reagents (Duolink PLA kit, No. DUO92101, Sigma-Aldrich), we were unable to complete the assay before the manuscript revision deadline. Consequently, we removed the PLA data and provided new evidence demonstrating the binding between endogenous Rab2A and HSD17B13 (Figure 4C). Lastly, our results indicated that HSD17B13 deficiency reduced Golgi-lipid droplet contacts, subsequently impairing VLDL secretion (Fig. 4D-K).

The authors implicate SNAREs in their proposed mechanism. I find this poorly justified and, given the novelty of this proposed mechanism, weakly evidenced. In a way, this alone would be significant enough for a publication, but in its current form, there are many other interpretations of the data. SNAREs are thought to mediate lipid droplet fusion, so it is already known that they localize there, and the mechanistic insight into the final model is, to my mind, not present.

Response: Thanks for your comments, as you pointed out, our studies on SNARE complex are not sufficiently rational and solid. Given the limited time for revision, we have carefully considered your suggestions and decided to remove the section on SNARE proteins in the revised manuscript.

Minor

I think showing a full gel for the Rab2A KO is crucial, as a single exon is deleted. A truncation protein may be made, which would not affect any interpretations but is important for future experimenters.

Response: The full gel proving Rab2A knockout in LCK mouse is shown in Appendix Figure S1C. The result predicts that no potential truncation proteins are present.

"Intriguingly, the Golgi localization of Rab2A appears largely contingent upon its activation state; GTP-bound Rab2A, signifying active status, preferentially localizes to the Golgi, as shown by overexpressing a constitutively active Rab2A mutant (Q65L)." - I am not sure why this is intriguing, as it is exactly how one would expect a Rab to function. Activated Rabs are recruited to membranes.

Response: Thanks for your comments. Actually, previous studies have shown that Rab2A is primarily located at Golgi apparatus, as indicated by staining with endogenously genomic-labeled Rab2A (PMID:33822845, 29940804) and exogenously overexpressed Rab2A (PMID:25453831, 30957628). Our findings corroborate these results, demonstrating that GTP-bound Rab2A is predominantly localized to the Golgi apparatus rather than the ER or ER-Golgi intermediate compartment (ERGIC) compartment, as confirmed by co-localization studies of

exogenously expressed Rab2A (Appendix Fig. S5). Further analysis with endogenous Rab2A also confirmed its predominant localization to the Golgi apparatus (Fig.3A). We revised the writing in the paper.

Rab2A's localization to the Golgi is not novel; however, Rab2A has been reported to localize to the ERGIC. The localization quantification in this figure is substandard, and proper repeats and quantification should be undertaken.

Response: Please refer to the above response. We repeated the assay and quantified the colocalization as shown in Appendix Figure S5 and Figure 3A.

"These experiments unveiled that fasting markedly dampened Rab2A activity, subsequently reducing its association with Golgi (Fig. 3H, I)" - Again, the controls showing the purification of the compartment are important. In addition, this assay does not directly show Rab2A activity and should not be reported as such.

Response: First, the method for Golgi apparatus purification employed in our revised manuscript is widely used (PMID: 32103509, 22661308, 12960170) and has been validated (Appendix Fig. S4). The activity of Rab2A was assessed using a GRASP55 pulldown assay, which specifically binds to the GTP-bound form of Rab2A. Our results indicated that fasting suppressed Rab2A activity, likely via AMPK signaling, thereby reducing its association with the Golgi apparatus (Fig. 5A).

"Rab2A may act as a tethering molecule" - I understand the authors' intention here. However, this is perhaps worth rewording, as Rab2A will recruit a number of effectors, one of which may be involved in tethering.

Response: We have revised the description in the revised manuscript accordingly.

Perhaps I am wrong; however, I suspect the authors have used ChatGPT to assist with the writing of this manuscript. I have no problem with this in principle as a reviewer, as on the whole the manuscript is clear, however there are number of sections where the writing needs to be modified. Particularly in the figure legends where phrases such

as "Extensive assays were conducted", "The graphical abstract delineates", "schematic depiction elucidates", and "The binding efficiency was meticulously assessed step by step using Flag affinity or MYC-affinity beads pulldown assays."

Response: Thanks for your kind suggestions, we have revised the corresponding sentences and furthermore the whole manuscript.

Referee #3:

The manuscript titled "Golgi-lipid droplet contact supports very-low-density lipoprotein secretion in hepatocytes" presents a study on lipid delivery from lipid droplets to the Golgi apparatus, modulated by Rab2A. This research aims to elucidate the molecular mechanisms underlying VLDL secretion in hepatocytes. The initial observations show that Rab2A depletion results in decreased serum triglycerides (TG) and total cholesterol (TC) levels and increased serum APOB levels. Further investigation reveals that these decreased lipid levels are due to reduced VLDL release rather than uptake (tyloxapol treatment). Intracellularly, overexpressed Rab2A-GFP localizes to the Golgi, and its deactivation (A769662 treatment) leads to Rab2A dissociating from the Golgi. Using immunofluorescence, protein ligation assays, and immunoprecipitation, the authors tested the hypothesis that Rab2A interacts with lipid droplets via activated Rab2A's interaction with HSD17B13. The paper provides evidence that Rab2A-liver KO mice have impaired VLDL release, potentially due to disrupted lipid delivery from lipid droplets to the Golgi, where APOB lipidation occurs. However, several critical issues compromise the clarity and impact of the findings.

First, the rationale for selecting Rab2A as a central player in lipid droplet-Golgi interactions is inadequately justified. The manuscript lacks a comprehensive introduction to Rab2A's known functions and its significance in lipid metabolism. The study's foundation appears weak without a clear explanation of why Rab2A was chosen for investigation. The authors should provide a thorough background on Rabs and SNARE proteins, detailing their roles in intracellular trafficking and lipid metabolism. This would help readers understand the relevance of Rab2A in the context of VLDL secretion.

Response: We have revised the description on LD-Golgi contact, VLDL and Rab2A in the introduction section. Firstly, previous studies on LD-organelles contacts suggest the potential LD-Golgi contact in hepatocytes. Subsequently, VLDL₂ lipidation in the Golgi apparatus requires lipid transport from LDs, indicating a likely contact between

the Golgi and LDs. Finally, our prior researches demonstrated that Rab2A, a Golgi-resident protein, regulated serum lipid levels. These above backgrounds suggest a probable hypothesis that Rab2A regulates LD-Golgi contacts, thereby supporting VLDL secretion in hepatocytes. Our studies in this manuscript began with identifying the roles of Rab2A in regulating serum lipid levels and VLDL secretion. Then, we demonstrated the deficient lipids accumulation and VLDL₂ lipidation in Golgi apparatus. Finally, results showed that the binding between Rab2A and HSD17B13 facilitated LD-Golgi contact, and that AMPK signaling potentially suppressed the Rab2A-HSD17B13 interaction, LD-Golgi organelle contact, and subsequent VLDL secretion.

As suggested by other reviewers, we have removed the studies on the SNARE complex in the revised manuscript after our careful consideration.

Furthermore, the study relies heavily on overexpression systems, which can lead to artefactual results that do not accurately represent physiological conditions. Overexpression of proteins, such as Rab2A and SNAREs, can result in non-physiological interactions and localization, potentially skewing the study's conclusions. The authors should address the limitations of using overexpression models and provide data from endogenous protein levels to strengthen their claims.

Response: Thank you for this suggestion. In the revised manuscript, the subcellular localization of Rab2A (Fig. 3A, B) and HSD17B13(Fig. 4B), the interaction between these two proteins (Fig. 4A), and their contact points (Fig. 4C) were all confirmed using primary antibodies targeting the endogenous proteins.

The immunofluorescence images are difficult to interpret, and the rationale behind the experiments is inadequately explained. The use of different cell lines in different experiments adds confusion, and the data does not consistently support the claims or the manuscript title. The interaction has not been demonstrated for endogenously expressed proteins, nor is there evidence for the endogenous STX5-GS28-GS15-VAMP4 SNARE complex or endogenous Rab2A or VAMP4

interaction with lipid droplets. It is also puzzling how transmembrane VAMP4 can be associated with lipid droplets.

Response: Thanks for your comments. In the revised manuscript, we repeated massive immunofluorescence assays and further quantified the images. HEK293T cells were generally used for protein binding assays, while Huh7 cells or primary hepatocytes were applied for immunofluorescence assays. The specific cell lines used for each assay are well noted in the corresponding figure legends, and we have revised the rationales of corresponding experiments in manuscript. Additionally, based on suggestions from other reviewers, we have removed the studies on the SNARE complex in the revised manuscript after careful consideration.

In summary, despite extensive data, the manuscript fails to provide significant mechanistic insights into the Golgi's role in lipid droplet biogenesis.

Response: As described in the revised manuscript, our major contributions lie in identifying the roles of Rab2A in LD-Golgi contacts and elucidating the significant functions of these contacts in VLDL lipidation and secretion. Future studies will further explore the roles of the Golgi in lipid droplet biogenesis. Meanwhile, other groups have demonstrated the important functions of the Golgi apparatus in regulating LD-related proteins, including PNPLA3 and ATGL (PMID: 38657050, 32184397).

Specific Comments:

Title: The manuscript title is too broad. A more specific title reflecting the findings would be better.

Response: We revised the title to “Rab2A-dependent Golgi-Lipid droplet contact supports very-low-density lipoprotein secretion in hepatocytes” as you suggested.

Rationale for Rab2A: The authors should elaborate on the choice of Rab2A as an interacting partner of lipid droplets, as this is a crucial aspect of their research. The absence of a description of Rabs and SNAREs is a notable gap. The rationale for selecting Rab2A should be adequately explained in the Introduction.

Response: Thanks for your suggestions. We have revised the description of Rabs in the Introduction.

Knockout Models: No evidence is provided that the phenotype in knockout models could be rescued by reintroducing Rab2A.

Response: Given the limited revision time, we did not repeat the assay in LCK mice after a discussion with the editor. Meanwhile, our studies in primary hepatocytes demonstrated that overexpression of wild-type Rab2A in LCK hepatocytes can potentially rescue the VLDL secretion (Fig. EV4D, E).

Figure 1: The discrepancy in units used in Figure 1A, 1B (mg/dl) and Figure 1G, H (mmol/ul) for TG and TC needs clarification. Additionally, the rescue experiments should be more pronounced and better explained.

Response: The kits for TG and TC detection were procured from different suppliers, resulting in varying units for the standard samples. These details have been carefully described in the Methods section.

Figure 3: The authors should clarify the novelty of their finding that GTP-bound Rab2A is primarily located in the Golgi, as this has been previously demonstrated (PMID: 25377857, PMID: 30957628).

Response: Thanks for your suggestions, we have repeated the assay, and actually, Rab2A is primarily located in the Golgi (Appendix Fig. S5; Fig. 3A, B; Fig. EV2), the conclusion was also indicated by staining with endogenously genomic-labeled Rab2A (PMID:33822845, 29940804) and exogenously overexpressed Rab2A (PMID:25453831, 30957628).

Golgi Morphology: In Figure 3E and 3G, the authors concluded that Golgi morphology is not affected in LCK. However, the electron microscopy images suggest otherwise. Quantification of the Golgi morphology should be provided.

Response: We have repeated the assay and quantified the Golgi morphology

(Appendix Figure S6).

AMPK Agonist: The use of A769662 to reduce Rab2A activity and its effects on Golgi morphology should be better quantified and explained. The Rab2A mislocalization due to overexpression needs further clarification.

Response: Thanks for your suggestions, we have repeated and quantified the assay, demonstrating that suppression of Rab2A activity with A769662 induce a dispersed distribution of Rab2A (Fig. EV5A-C). Subsequently, our studies further revealed the partial re-localization of Rab2A from Golgi to lysosome upon inactivation (Fig. EV5D, E), contributing to phagosome formation, a phenomenon also supported by previous findings (PMID:30957628, 28424218).

Figures 4A and 4B: The authors should provide separate graphs for Golgi and ER TC and TG levels. The error bars for ER are challenging to see, and quantification for Figure 4C is missing.

Response: Thanks for your kind suggestions, we have revised the Figure (Figure 3C-E).

Figure 4D: Only one cell image showing colocalization is insufficient to demonstrate the interaction between Rab2A and lipid droplets. Colocalization studies and quantification are necessary.

Response: The localization of Rab2A at the Golgi apparatus was confirmed and quantified (Appendix Fig. S5; Fig. 3A, B). We also repeated and demonstrated the contacts between Golgi-localized Rab2A and LDs in Huh7 cells and primary hepatocytes (Fig. EV2). Additionally, further assays were conducted to investigate the roles of Rab2A in mediating Golgi-LD contacts (Fig. 3H-J).

Figure 4E and 4F: The lipid droplet pulldown experiment lacks detailed methodology and evidence. The experiment does not adequately explain the Golgi-LD interface.

Response: Thanks for your suggestions, we have revised the methods for the lipid

droplet pulldown experiment in Methods section.

Figure 5: The overexpression of proteins in the experiments raises concerns about the findings. Colocalization statistics are missing, and the rationale for using RCAS1-BFP for Golgi labeling is unclear.

Response: Thanks for your suggestions. We initially implied overexpression system to prove the binding efficiency between Rab2A and HSD17B13 (Fig. EV3). Further we quantified the percentage of HSD17B13 in contact with Rab2A (Fig. EV3D, E). In addition, we also evaluated the binding and colocalization between endogenous Rab2A and HSD17B13 (Fig. 4A-C). We selected RCAS1-BFP protein for Golgi labeling in some assays due to its convenience, especially in four-channel immunofluorescence.

Golgi Fragmentation: The fragmented Golgi in Figure 5F and EV3 needs further investigation and explanation.

Response: Studies in fly larval ventral nerve cords (PMID: 33822845) and HeLa-S3 cells (PMID: 26209634) have highlighted Rab2A's role in maintaining Golgi apparatus integrity. Our investigations extend these findings to primary hepatocyte, indicating that Rab2A deficiency has a negligible impact on the structural integrity of the Golgi apparatus (Appendix Fig. S6), suggesting varied roles of Rab2A in different cell types.

Figure 6B: The explanation of PLA blobs is inadequate. The differences between treated and untreated cells are difficult to discern from the IF images.

Response: Thanks for your kind suggestions. We initially intended to repeat the PLA assay to address concerns about the image quality. Unfortunately, the delivery time for the necessary reagents (Duolink PLA kit, No. DUO92101, Sigma-Aldrich) exceeded the manuscript's revision deadline. As a result, we removed the PLA data and instead provided new data to demonstrate the binding between endogenous Rab2A and HSD17B13 (Figure 4C).

VLDL Secretion: The authors claim that the Rab2A-HSD17B13 complex mediates VLDL secretion, but they provide no direct data supporting VLDL secretion.

Response: As detailed in the revised manuscript, we have provided evidence demonstrating that Rab2A and HSD17B13 regulate VLDL secretion in primary hepatocytes and mouse models (Fig. 2A; Fig. EV1A-D; Fig. 4G-K; Appendix Fig. 10). Additionally, we showed that disrupting the interaction between Rab2A and HSD17B13 with a truncated Rab2A protein failed to rescue VLDL secretion in LCK primary hepatocytes (Fig. EV4). While we acknowledge that the use of radiolabeled palmitate or oleic acid would directly quantify VLDL secretion, we regret that we do not have permission to use isotopes. Nonetheless, we believe that the presented data sufficiently supports our conclusions.

SNARE Proteins: The study on SNARE proteins lacks proper citations and rationale. Overexpression studies do not convincingly demonstrate SNARE interactions.

Response: Thanks for your comments. Considering suggestions from other reviewers and the limited time for revisions, we have carefully decided to remove the studies on the SNARE complex from the revised manuscript. However, the data presented in the revised manuscript adequately supports our conclusions. We plan to further investigate the SNARE complex and publish these findings in a future paper.

Figure 7B: The image quality is poor, and it is challenging to identify GS28 and VAMP4. The SNARE interaction claims require more robust evidence.

Response: Thanks for your comments. Considering suggestions from other reviewers and the limited time for revisions, we have carefully decided to remove the studies on the SNARE complex from the revised manuscript. However, the data presented in the revised manuscript adequately support our conclusions. We plan to further investigate the SNARE complex and publish those findings in a future paper.

GS28 Distribution: The increased Golgi localization of GS28 in LCK mice and its

implications for SNARE complex formation need clarification. The role of other SNARE partners should be discussed.

Response: Thanks for your comments. Considering suggestions from other reviewers and the limited time for revisions, we have carefully decided to remove the studies on the SNARE complex from the revised manuscript. However, the data presented in the revised manuscript adequately support our conclusions. We plan to further investigate the SNARE complex and publish those findings in a future paper.

Novel SNARE Complex: The claim of identifying a novel SNARE complex (Syntaxin-5/GS28/GS15/VAMP-4) is unsubstantiated. More robust data is needed.

Response: Thanks for your comments. Considering suggestions from other reviewers and the limited time for revisions, we have carefully decided to remove the studies on the SNARE complex from the revised manuscript. However, the data presented in the revised manuscript adequately support our conclusions. We plan to further investigate the SNARE complex and publish those findings in a future paper.

Model in Figure 8: The model explaining Rab2A's role in Golgi-LD contacts and VLDL secretion is poorly detailed. The discussion on VLDL1 and VLDL2 is missing.

Response: We have revised the graphical model.

Discussion: The discussion section needs significant improvement for clarity and coherence.

Response: We have revised the description in the Discussion section.

Dear Dr Chen,

Thank you submitting a revised version of your manuscript. It was sent to the same reviewers that originally appraised your work; we have now received their reports and attached them to the bottom of this email. As you will see, Referees #1 and #3 are satisfied with the changes you made (although some quantitation of the data presented is requested). Referee #2, however, remains unconvinced that the data you present are sufficiently compelling to define Rab2a-mediated 'contact sites'. On this point, I agree. I would therefore ask you to remove the term 'contact' from title and abstract. Please focus the description of your results on 'protein-protein' interaction. I think the discussion would be a good place for you to weigh the drawbacks and limitations of the techniques you have used, and (if you wish) suggest that Rab2a-HSD17B13 interaction defines the position of LD-Golgi contact sites.

Before we can move forwards towards publication of your manuscript, there are also some remaining editorial points which need to be addressed. In this regard, would you please:

- include the details of author Na Ni in our online submission system and provide institutional email addresses for co-corresponding authors: Yi-ning Yang (yangyn5126@163.com) and Shuai Chen (schen6@163.com),
- acknowledge funding in our online submission system from the Doctoral Start-up Foundation of Anhui Medical University (0810013101), the Outstanding Youth Science Foundation of the Department of Education of Anhui Province (2023AH030058), the Technology Innovation Leading Talent Project (2022195955), and the Ministry of Science and Technology of China (2018YFA0801100 and 2021YFF0702100),
- use the reference format of 10 authors + et al. instead of 20 + et al.,
- remove the author credit section from the manuscript,
- upload EV figures as individual, high-resolution figure files with EV figure legends included in the main manuscript file below main figure legends,
- provide the Appendix file in PDF format; nomenclature should be Appendix Figure S1-S12 and Appendix Table S1-S2 with the corresponding callouts; add page numbers to the Table of Contents,
- include a 'Reagents and Tools' table,
- provide exact p values in the legends of figures 1a-b, g-h, l; 2a; 3c-d, i; 4f-g, j-k; 5b, e, g; EV 1a, c, g-h; EV 4e; EV 5c, and
- correct the mismatch between the annotated p values in the figure legend and the annotated p values in the figure file figures EV 1a, c, g-h.

We require the publication of source data for electrophoretic gels and blots, with the aim of making primary data more accessible and transparent to the reader. Please therefore provide me with a PDF file per figure that contains the original, uncropped and unprocessed, high-resolution scans of all or key gels used in the figures. Please make sure data for figures EV3 and supplementary figure S8 are included. The PDF files should be labeled with the appropriate figure/panel number, and should have molecular weight markers; further annotation could be useful but is not essential. The PDF files will be published online with the article as supplementary "Source Data" files. Source Data can also include Excel tables to accompany your graphs. We anticipate that their inclusion will make your work more discoverable and useable to scientists in the future. Hannah Sonntag will contact you separately about the Source Data for this manuscript.

We include a synopsis of the paper (see <http://emboj.embopress.org/>). Please provide me with a two-sentence general summary statement and 3-5 bullet points that capture the key findings of the paper.

We also need a summary figure for the synopsis. The size should be 550 wide by [200-400] high (pixels). You can also use something from the figures if that is easier.

I look forward to receiving these changes. EMBO Press is an editorially independent publishing platform for the development of EMBO scientific publications.

Best wishes,

William

William Teale, PhD
Editor
The EMBO Journal
w.teale@embojournal.org

We realize that it is difficult to revise to a specific deadline. In the interest of protecting the conceptual advance provided by the work, we recommend a revision within 3 months (26th Dec 2024). Please discuss the revision progress ahead of this time with the editor if you require more time to complete the revisions. Use the link below to submit your revision:

Referee #1:

The authors have addressed all our concerns. We congratulate them on an interesting and timely study.

One minor point: several sentences are repeated verbatim between the introduction and discussion sections. These should be re-written to avoid redundancy and maximize usefulness for the reader.

Referee #2:

Xu et al. have resubmitted their manuscript and responded to the reviewer's comments. I appreciate the responses; several points have been addressed thoroughly. The authors have introduced new data, including new endogenous interaction data, and removed the SNARE data, which I think has improved the flow of the manuscript and removed potentially incomplete findings. Although I appreciate these efforts, some of my original concerns remain. I have listed these below:

1) The authors "removed the PLA data and provided new evidence demonstrating the binding between endogenous Rab2A and HSD17B13 (Figure 4C)."

Again, I consider this data weak evidence. There is a very small spot of partial colocalisation, and I see no controls or statistics. If the proteins do not colocalise (which is my interpretation of this data), the authors should state they do not rather than claim they do. There is no control (other Golgi proteins, such as organelle markers). Additionally, co-localisation by classical microscopy does not and cannot demonstrate a protein interaction.

2) Despite removing the SNARE data, the authors' ultimate interpretation here is that there is a contact site. However, there is simply very limited evidence of a contact site. Contact sites are well-defined and include several parameters, including a protein complex in juxtaposed lipid bilayers. The evidence presented here from light microscopy is simply not high enough resolution to demonstrate this.

I hugely support the author's work here, and combining cell biology with animal models is laudable. I also appreciate the

practical and precise response to the reviewers. However, I remain concerned with the scope of the claims of the manuscript based on the presented evidence and the complexity of the data. I also retain my concerns with the interpretation and the intention of the interpretation, for example the claims of colocalisation are simply without support and in fact, the opposite appears to be the case. There also remain several very minor points, for example figure 2A has quantification without a control, although the Golgi purification control was included it was not on the actual presented data for the claim (as far as I can tell), and the evidence of organelle contacts (Figure 5D) is very weak. On balance, with the remaining issues, I feel I can only recommend more revisions. It is my suggestion to the authors that they simplify the story and demonstrate some of the claims with solid evidence rather than many with what I consider weak evidence.

Referee #3:

The revised manuscript is greatly improved and more streamlined. The data clearly demonstrate that Rab2a interacts (directly or indirectly) with HSD17B13, and that disrupting this interaction-genetically or pharmacologically-affects VLDL secretion in hepatocytes. It is also evident that Rab2a is not essential for Golgi-LD contacts, as Rab2a depletion results in only a slight reduction in these contacts (from 22 to 17). However, some results still require quantification, specifically the binding of HSD17B13 to various Rab2a mutants (EV3F, S8A-C, S9) and the relocalization of Rab2a in cells treated with A769662.

Dear Editor William Teale,

We sincerely thank you and the reviewers for the valuable feedback and suggestions, which have significantly enhanced the rigor and quality of our manuscript. Specifically, we have recognized that the use of the term "Contact" in our study was not sufficiently precise. In response, we have carefully revised this terminology for greater accuracy, as recommended. Additionally, we have addressed all formatting issues and corrected the noted errors. Below, we provide a detailed point-by-point response to the reviewers' comments.

We look forward to your positive response.

Sincerely,

Liang Chen

Referee #1:

The authors have addressed all our concerns. We congratulate them on an interesting and timely study.

One minor point: several sentences are repeated verbatim between the introduction and discussion sections. These should be re-written to avoid redundancy and maximize usefulness for the reader.

Response: Thanks for your kind suggestions, we have carefully revised the sentences and descriptions throughout our manuscript and hope the changes meet with your approval.

Referee #2:

Xu et al. have resubmitted their manuscript and responded to the reviewer's comments. I appreciate the responses; several points have been addressed thoroughly. The authors have introduced new data, including new endogenous interaction data, and removed the SNARE data, which I think has improved the flow of the manuscript and removed potentially incomplete findings. Although I appreciate these efforts, some of my original concerns remain. I have listed these below:

1) The authors "removed the PLA data and provided new evidence demonstrating the

binding between endogenous Rab2A and HSD17B13 (Figure 4C).".

Again, I consider this data weak evidence. There is a very small spot of partial colocalization, and I see no controls or statistics. If the proteins do not colocalize (which is my interpretation of this data), the authors should state they do not rather than claim they do. There is no control (other Golgi proteins, such as organelle markers). Additionally, co-localization by classical microscopy does not and cannot demonstrate a protein interaction.

Response: Thanks for your comments. We have repeated the assay, now including the staining of the Golgi and Lipid droplets (Figure 4C). Additionally, we have also revised the corresponding sections in the manuscript as suggested. We think all the relevant data presented sufficiently support the conclusion of binding and partial colocalization between Rab2A and HSD17B13.

2) Despite removing the SNARE data, the authors' ultimate interpretation here is that there is a contact site. However, there is simply very limited evidence of a contact site. Contact sites are well-defined and include several parameters, including a protein complex in juxtaposed lipid bilayers. The evidence presented here from light microscopy is simply not high enough resolution to demonstrate this.

Response: Thanks for your comments, and we agree that the evidence presented in our study is not yet sufficient to conclusively demonstrate organelles contacts. In the revised manuscript, we have refined the description to focus on Golgi-LD interactions, we will do more work to clarify the underlying mechanisms in future studies.

I hugely support the author's work here, and combining cell biology with animal models is laudable. I also appreciate the practical and precise response to the reviewers. However, I remain concerned with the scope of the claims of the manuscript based on the presented evidence and the complexity of the data. I also retain my concerns with the interpretation and the intention of the interpretation, for example the claims of colocalization are simply without support and in fact, the opposite appears to be the case. There also remain several very minor points, for

example figure 2A has quantification without a control, although the Golgi purification control was included it was not on the actual presented data for the claim (as far as I can tell), and the evidence of organelle contacts (Figure 5D) is very weak. On balance, with the remaining issues, I feel I can only recommend more revisions. It is my suggestion to the authors that they simplify the story and demonstrate some of the claims with solid evidence rather than many with what I consider weak evidence.

Response: Thanks for your valuable suggestions. Upon your recommendations, we have revised several data and descriptions in our manuscript to enhance clarity and simplicity (Appendix Figure S13; Figure 5D, 4E, 3H). We hope these changes meet your approval.

Referee #3:

The revised manuscript is greatly improved and more streamlined. The data clearly demonstrate that Rab2a interacts (directly or indirectly) with HSD17B13, and that disrupting this interaction-genetically or pharmacologically-affects VLDL secretion in hepatocytes. It is also evident that Rab2a is not essential for Golgi-LD contacts, as Rab2a depletion results in only a slight reduction in these contacts (from 22 to 17). However, some results still require quantification, specifically the binding of HSD17B13 to various Rab2a mutants (EV3F, S8A-C, S9) and the re-localization of Rab2a in cells treated with A769662.

Response: Thanks for your kind comments and suggestions. We have quantified almost all the data related to the binding between Rab2A and HSD17B13 in our revised manuscript. We hope these revisions meet your approval.

Dear Dr. Chen,

In my last letter, I asked for Source Data. Thank you for providing these for the main figures. However, in my letter, I asked you to please include original, unmodified images for figures EV3 and supplementary figure S8. Please send your revised version of this manuscript in a submission that includes these images.

Best wishes,

William Teale

William Teale, PhD
Editor
The EMBO Journal
w.teale@embojournal.org

All editorial and formatting issues were resolved by the authors.

Dear Dr. Chen,

I am pleased to inform you that your manuscript has been accepted for publication in the EMBO Journal.

Congratulations to you and your lab on the publication of a really exciting piece of work!

Yours sincerely,

William Teale

William Teale, PhD
Editor
The EMBO Journal
w.teale@embojournal.org
